# Dual mechanism β-amino acid polymers promoting cell adhesion

Qi Chen [1], Donghui Zhang [2], Wenjing Zhang [2], Haodong Zhang [2], Jingcheng Zou [2], Mingjiao Chen [3], Jin Li[3], Yuan Yuan[2] & Runhui Liu [1,2✉]

Cell adhesion has tremendous impact on the function of culture platforms and implants. Cell-adhesive proteins and peptides have been extensively used for decades to promote cell adhesion, however, their application suffers from their easy enzymatic degradation, difficulty in large-scale preparation and expensiveness. To develop the next-generation cell-adhesive materials, we mimic the cell adhesion functions and mechanisms of RGD and KRSR peptides and design cell-adhesive cationic-hydrophobic amphiphilic β-amino acid polymers that are stable upon proteolysis and easily prepared in large scale at low cost. The optimal polymer strongly promotes cell adhesion, using preosteoblast cell as a model, by following dual mechanisms that are independent of sequence and chirality of the statistic copolymer. Our strategy opens avenues in designing the next-generation cell-adhesive materials and may guide future studies and applications.

[1] State Key Laboratory of Bioreactor Engineering, East China University of Science and Technology, Shanghai 200237, China. [2] Key Laboratory for Ultrafine Materials of Ministry of Education, Frontiers Science Center for Materiobiology and Dynamic Chemistry, Research Center for Biomedical Materials of Ministry of Education, School of Materials Science and Engineering, East China University of Science and Technology, Shanghai 200237, China. [3] Shanghai Key Laboratory of Orbital Diseases and Ocular Oncology, Department of Ophthalmology, Ninth People's Hospital, Shanghai Jiao Tong University School of Medicine, Shanghai 200011, China. ✉email: rliu@ecust.edu.cn

Cell adhesion greatly affected the cell fate, such as cell signaling, migration, proliferation and differentiation[1–3]. Favorable cell adhesion is critical in tissue engineering and biomedical study and application for its tremendous impact on the function of culture platforms and implants[4–7]. Cell-adhesive proteins from extracellular matrix (ECM), such as fibronectin[8], have been extensively used to promote cell adhesion. However, these naturally derived materials have concerns of infections and immunogenicity and are hard to obtain in large quantity[9]. Hence, ECM protein-derived cell-adhesive peptides, such as the gold standard Arg-Gly-Asp (RGD) peptide, are extensively used as the substitutions of ECM proteins in promoting cell adhesion[10–13]. Nevertheless, the application of cell-adhesive peptides suffers from their susceptibility to proteolysis, high price, and time-consuming synthesis. Therefore, it is in great need to develop cell-adhesive materials that are stable upon protease, cheap and easy to prepare in large quantity.

To develop the next-generation cell-adhesive materials, we turn our attention to β-amino acid polymers that have a polyamide backbone similar to natural peptides and can be easily synthesized in large scale and at low cost via the ring-opening polymerization of β-lactams[14–18]. Inspired by the success of mimicking antimicrobial host defense peptides using statistic copolymers[14,15,19–22], we designed cell-adhesive β-amino acid polymers by mimicking natural cell-adhesive peptides because the slow progress of bone regeneration and implant integration especially suffers from the quick degradation of prefunctionalized cell-adhesive proteins/peptides and loss of functions in vivo. We also designed the cell-adhesive β-amino acid polymers by incorporating two cell-adhesive mechanisms as is widely used in dual-ligand systems to greatly enhance cell adhesion[23,24]. The Lys-Arg-Ser-Arg (KRSR) peptide enables osteoblast-selective cell adhesion via its binding to cell surface heparin that has negative charges[25]. The high proportion of positive charge within KRSR, three out of four amino acids having positive charges, implies the importance of positive charges for preosteoblast adhesion peptide. Implant biomaterials will unavoidably contact ECM proteins and serum proteins[26], which inspired us to utilize these host proteins to acquire cell adhesion.

To find cell-adhesive materials, using preosteoblast adhesion as the model for proof-of-concept demonstration, we design cationic-hydrophobic amphiphilic β-amino acid polymers. We introduce positive charges to mimic KRSR for binding with cell surface heparin and possible other negatively charged polysaccharides; the amphiphilic polymer structure with tunable cationic/hydrophobic subunit ratio will facilitate adsorption of host proteins that possess RGD motifs to bind cell surface integrins and enable cell adhesion. The β-amino acid polymers display superior performance in preosteoblast adhesion by employing dual mechanisms of cell adhesion, the cell surface integrin-dependent and polysaccharide-dependent mechanisms (Fig. 1a). Moreover, the β-peptide backbone provides these polymers favorable biocompatibility and in vivo stability. Our study suggests avenues in designing cost-effective and in-vivo-stable cell-adhesive materials for tissue regeneration and permanent implants.

## Results

### High-throughput screening of β-amino acid polymers for preosteoblast cell adhesion.
To generate a library of cationic-hydrophobic amphiphilic β-amino acid polymers, we incorporated a racemic hydrophobic subunit: CP (cyclopentyl), CH (cyclohexyl), or CO (cyclooctyl) and a racemic cationic subunit: NM ("no methyl"), MM ("monomethyl"), DM ("dimethyl"), or HLys (homo-lysine) into the polymer chain with incrementally

increased ratio of the cationic subunit, from 40% to 100% (Fig. 1b, c and Supplementary Fig. 1). These heterochiral β-amino acid polymers were conveniently synthesized from the one-pot ring-opening polymerization of β-lactams to have an average length of ~20 subunits and an N-terminal thiol group that enabled easy modification of these polymers to maleimide-functionalized glass slides for high-throughput screening on preosteoblast cell adhesion (Fig. 1c–e). An OEG8 antifouling layer was used between β-amino acid polymers and glass surface to reduce the non-specific cell adhesion from surface fouling and to demonstrate the genuine cell-adhesive function of these polymers, as demonstrated in our recent study[27]. The capability of β-amino acid polymers for cell adhesion and growth was examined by measuring the fluorescence intensity of live/dead staining on preosteoblast cells that were cultured for 2 days on variable β-amino acid polymer-modified surfaces, using cell adhesion gold standard RGD peptide (RGDSPC) and preosteoblast-selective KRSR peptide (KRSRGYC) for comparison (Fig. 1f). The subunit composition within polymer chains dominated polymers' function and multiple polymers displayed excellent cell adhesion, with DM:CO series at 1:1 subunit ratio ($DM_{50}CO_{50}$) demonstrating the best performance in promoting preosteoblast adhesion and growth, comparable to the gold standard RGD peptide and superior to the KRSR peptide (Fig. 1g, h; $^1$H NMR spectrum and gel permeation chromatography characterization of DM:CO polymers in Supplementary Figs. 2 and 3, respectively). The optimal polymer, $DM_{50}CO_{50}$, also showed favorable aqueous solubility and was selected for surface characterization and continuous studies.

X-ray photoelectron spectroscopy (XPS) characterization showed apparent C1$s$ and N1$s$ peaks after the amination step, indicating a successful surface functionalization with amine groups via (3-aminopropyl) triethoxysilane (APTES) modification (Fig. 1i). A subsequent modification of OEG8 antifouling layer and further $DM_{50}CO_{50}$ functionalization were confirmed by the change of C:N element ratio (Supplementary Table S1). The 3D atomic force microscopy (AFM) characterization showed that surfaces modified with either RGD or $DM_{50}CO_{50}$ have increased $R_a$ value, a value that reflects the surface roughness, compared to bare glass surface (Fig. 1j). It is noteworthy that $DM_{50}CO_{50}$ has homogeneous modification on the surface, which is favorable to generate reproducible and homogenous protein adsorption and subsequent cell adhesion. Water contact angle measurement showed that after the bare glass surface was modified with OEG8 and $DM_{50}CO_{50}$ sequentially, the surface water contact angle changed from 57° to 44° and then to 57°, which is consistent to the amphiphilic property of $DM_{50}CO_{50}$ compared to the more hydrophilic RGD that gave modified surface water a contact angle of 41° (Fig. 1k).

### Cell adhesion and proliferation on surfaces modified with $DM_{50}CO_{50}$, RGD, and KRSR.
To evaluate the optimal β-amino acid polymer, $DM_{50}CO_{50}$, in detail for its ability to support preosteoblast cell adhesion and cell proliferation, we characterized cell morphology on polymer-modified surfaces using fluorescence confocal microscopy after cells were stained for actin (green), vinculin (red), and nucleus (blue). In general, 3 h after cell seeding, cell spreading areas on $DM_{50}CO_{50}$ surface and KRSR surface were lower than that on RGD surface; after 24 h, cell spreading areas on $DM_{50}CO_{50}$ surface and RGD surface are comparable, and both are significantly better than cell spreading on KRSR surface (Supplementary Fig. 4–6). Watching more closely, we found that 3 h after cell seeding preosteoblast cells on RGD peptide-modified surface already formed polygonal and highly spreading morphology, with well-defined actin stress fibers

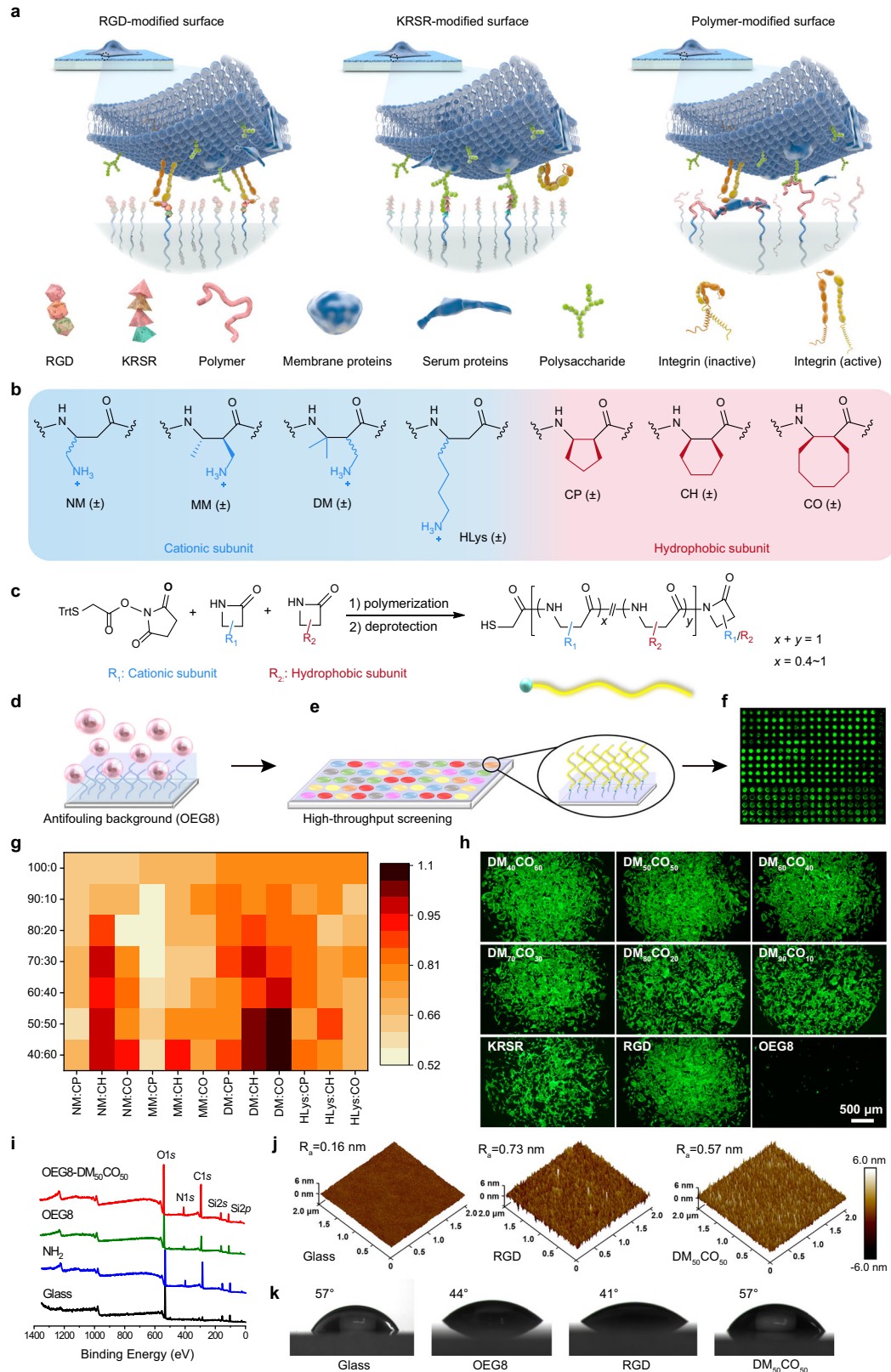

and focal adhesions; cells on DM$_{50}$CO$_{50}$ and KRSR peptide-modified surfaces formed punctuate focal adhesions and randomly oriented stress fiber. Cell spreading area on DM$_{50}$CO$_{50}$-modified surface fell between that on RGD and KRSR-modified surfaces (Fig. 2a, c and Supplementary Fig. 4). After 24 h culture, cells showed similar morphology on DM$_{50}$CO$_{50}$ and RGD peptide-modified surfaces, fully spread and with many bundles of actin stress fibers and extended focal adhesion, which is distinct from the narrow arrangement of actin microfilaments and less spread cell morphology on KRSR-modified surfaces (Fig. 2b, c and Supplementary Fig. 5). The overall performance of DM$_{50}$CO$_{50}$ implied a cell adhesion mechanism possibly different

**Fig. 1 High-throughput screening of β-amino acid polymers for preosteoblast cell adhesion. a** Cell adhesion mechanisms of RGD, KRSR, and β-amino acid polymer-modified surfaces. **b** Subunits to compose β-amino acid polymers. All subunits are racemic and the resulting polymers are heterochiral. **c** The general synthesis of the 76 β-peptide polymers with a chain length of 20 amino acid residue. **d** An OEG8 antifouling layer was used to resist protein adsorption and non-specific cell adhesion to the substrate. **e** Thiol-terminated polymers were covalently attached to maleimide-functionalized glass surface for high-throughput screening of cell adhesion, using RGD (RGDSPC) and KRSR (KRSRGYC) peptide-modified surfaces for comparison. **f** High-throughput screening of cell adhesion evaluated by fluorescence scanning and quantification on live/dead stained cells, calcein AM (green) for live cells. **g** Heat map showing the relationship of preosteoblast adhesion vs. polymer composition. The fluorescence intensity was normalized with RGD-modified surface. **h** Micrograph of F-actin (green) stained preosteoblast cell adhesion to surfaces modified with $DM_xCO_y$ ($x + y = 100$, $x = 40, 50, 60, 70, 80, 90$), RGD, KRSR, and OEG8, respectively. Scale bar: 500 μm. **i** XPS spectra of bare glass, $NH_2$-glass, OEG8, and $DM_{50}CO_{50}$-modified surfaces. **j** 3D AFM images of bare glass, RGD, and $DM_{50}CO_{50}$-modified surfaces. **k** Water contact angle of bare glass, OEG8, RGD, and $DM_{50}CO_{50}$-modified surfaces.

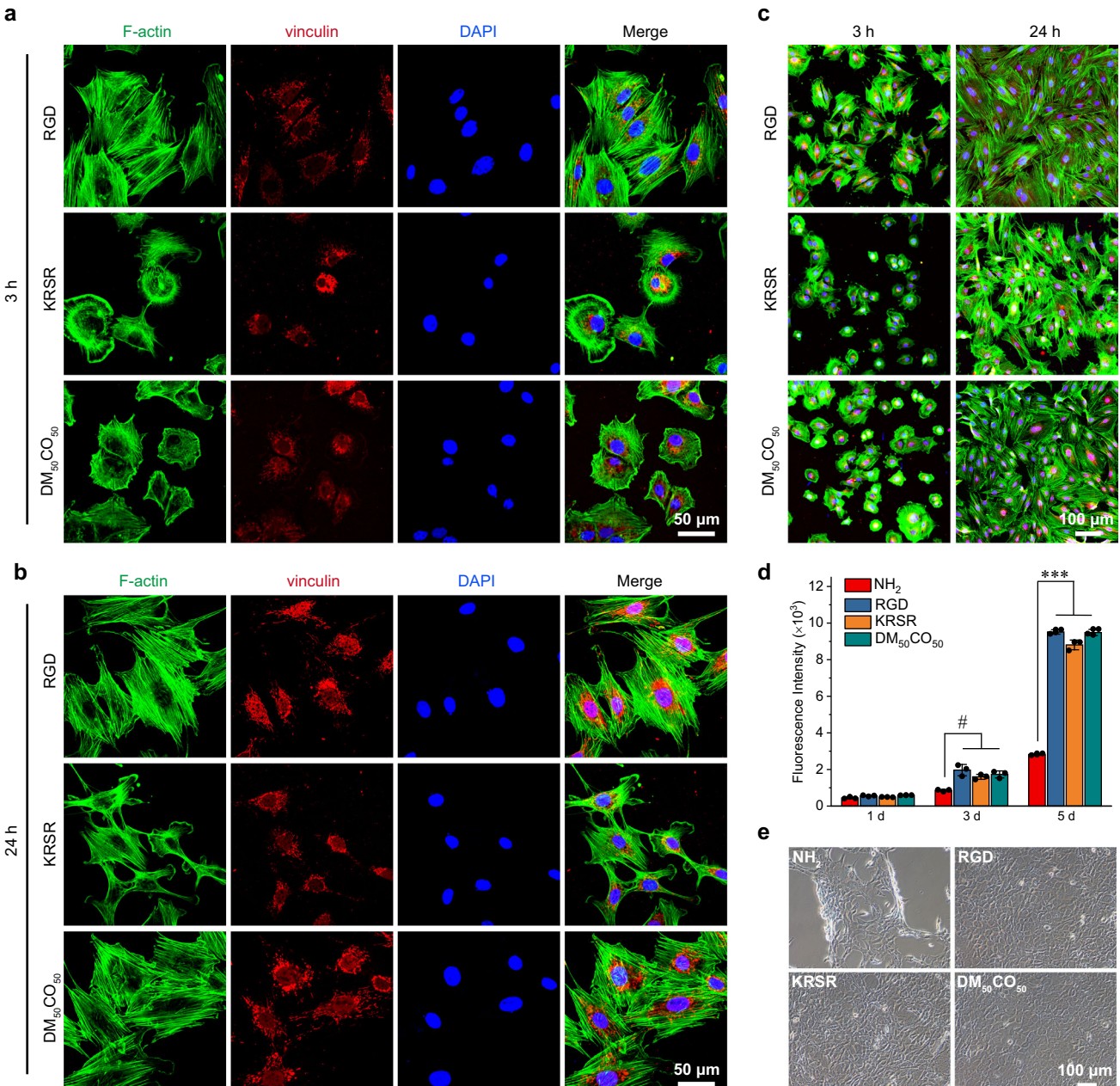

**Fig. 2 Adhesion and proliferation of preosteoblast cells. a–c** Fluorescence confocal microscope images (green, actin; red, vinculin; blue, nucleus) of preosteoblast cells on RGD, KRSR, and $DM_{50}CO_{50}$-modified surfaces after cell seeding for 3 h (**a**) and 24 h (**b**), Scale bar: 50 μm. Lower magnification images (**c**) after cell seeding for 3 and 24 h, Scale bar: 100 μm. **d** Cell proliferation after 1, 3, and 5 days of culture. Quantification of cell density was done with Alamar Blue assay. **e** Bright field images of preosteoblast cells cultured on $NH_2$, RGD, KRSR, and $DM_{50}CO_{50}$-modified surfaces for 5 days. Scale bar: 100 μm. Data (in **d**) represent mean ± s.d. ($n = 3$). Statistical analysis: one-way ANOVA with Tukey post-test, $^\#p < 0.05$, $^{***}p < 0.0001$.

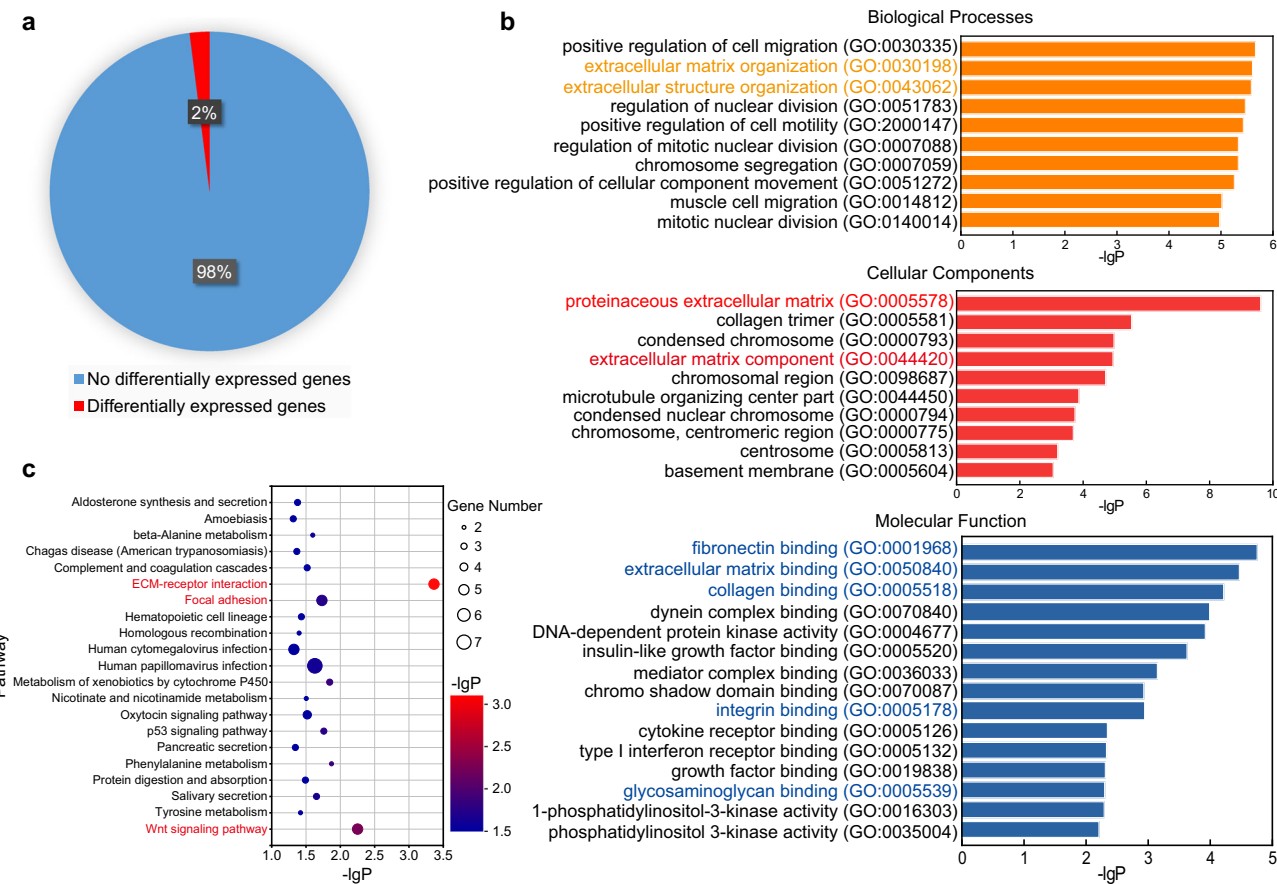

**Fig. 3 RNA-seq analysis on DM$_{50}$CO$_{50}$-modified surface compared to RGD-modified surface. a** From 17,267 gene transcripts in analysis, 305 differentially expressed genes showed greater than 50% change in expression level after 2 days of culture. **b** Significant GO terms of associated biological processes, cellular components, and molecular functions from differentially regulated genes. **c** KEGG pathway enrichment. The y-axis label represents KEGG pathway, and the x-axis shows -lgP. The size of the dots represents the number of gene enrichment.

from that of RGD peptide and KRSR peptide. Cell proliferation study up to 5 days showed that DM$_{50}$CO$_{50}$ can promote pre-osteoblast cell proliferation as good as RGD peptide and KRSR peptide do, and substantially better than the amine-functionalized cell-adhesive surface (Fig. 2d, e and Supplementary Fig. 7).

**RNA-seq analysis of cell adhesion on DM$_{50}$CO$_{50}$-modified surfaces**. To figure out the mechanism of DM$_{50}$CO$_{50}$ in promoting cell adhesion, we compared gene expression in cells that adhered to surfaces modified with RGD peptide and DM$_{50}$CO$_{50}$. From 17,267 gene transcripts, we identified total 305 differentially expressed genes (DEGs) that showed greater than 50% change in expression level, representing 2% of total genes in analysis after 2 days of culture (Fig. 3a). To identify the relation of these DEGs to key biological processes and pathways, we performed gene ontology (GO) enrichment analysis[28] on three GO categories, biological processes, cellular components, and molecular functions (Fig. 3b). The analysis showed significant enrichment of 171 GO terms ($p < 0.05$, $p$ represents the significance level of enrichment), including 136 for biological processes, 22 for cellular components, and 13 for molecular functions. We picked the top 10 or 15 key GO terms in each GO category according to the significance ($p$ value) and found multiple ECM-protein-related and significantly enriched GO terms that highlight the widespread effect of RGD vs. DM$_{50}$CO$_{50}$ on preosteoblast cell, including: ECM organization (GO:0030198) and extracellular structure organization (GO:0043062) in biological processes,

proteinaceous ECM (GO:0005578) and ECM component (GO:0044420) in cellular components, and fibronectin binding (GO:0001968), ECM binding (GO:0050840), collagen binding (GO:0005518), and integrin binding (GO:0005178) in molecular functions. Overall, these terms related to the three GO categories indicated the difference between DM$_{50}$CO$_{50}$ and RGD peptide for their interactions with preosteoblast cells, from the downstream effect of ECM reorganization.

Among the top 10 key GO terms in biological processes, we also found four top GO terms that are related to cell motility and cell migration including positive regulation of cell migration (GO:0030335), positive regulation of cell motility (GO:2000147), positive regulation of cellular component movement (GO:0051272), and muscle cell migration (GO:0014812). The other four of the top 10 key GO terms in biological processes are related to cell division including regulation of nuclear division (GO:0051783), regulation of mitotic nuclear division (GO:0007088), chromosome segregation (GO:0007059), and mitotic nuclear division (GO:0140014). These results may imply that DM$_{50}$CO$_{50}$ polymer could influence cell adhesion as well as the migration capacity of preosteoblast cells and their capacity to colonize new areas during bone regeneration.

We also did Kyoto Encyclopedia of Genes and Genomes (KEGG) pathway enrichment analysis on DEGs and identified several significant signal transduction pathways (Fig. 3c). The Wnt signaling pathway and focal adhesion pathway are also present with a relatively high enrichment. We paid attention to

these two pathways that are known to involve in cell migration and cell proliferation. Focal adhesion involves in reorganization of the actin cytoskeleton and affects cell motility, cell proliferation, and cell survival. Wnt signaling pathway involves in remodeling of the cytoskeleton and changes in cell adhesion and motility. This result suggested that $DM_{50}CO_{50}$ polymer may not only supported the cell adhesion and spreading of preosteoblast, but also affect the mobility and proliferation of cells, which echoes the result of GO enrichment analysis above.

It is noteworthy that the most significant signal transduction pathway among 21 KEGG pathways ($p$ value < 0.05) is the ECM–receptor interaction (Fig. 3c), which is related to many terms in GO enrichment analysis, such as fibronectin binding and integrin binding. This result implied that $DM_{50}CO_{50}$ was different from RGD peptide in promoting preosteoblast cell adhesion via the ECM-related interactions with cell surface receptors. Because RGD peptide promoted cell adhesion via its direct binding to cell surface integrin, we continued to explore and compare the role of integrin-associated cell adhesion mechanism between RGD peptide and $DM_{50}CO_{50}$.

**Integrin-associated cell adhesion**. The integrin–RGD binding is divalent-cation-dependent and can be blocked by ethylenediaminetetraacetic acid (EDTA), a divalent cation chelator[29]. The presence of EDTA led to substantially diminished cell adhesion to RGD peptide-modified surface for both cell spreading area and cell density, in both serum-containing (Fig. 4a–c) and serum-free environments (Fig. 4d–f); whereas, EDTA resulted in moderate reduction of cell spreading area on $DM_{50}CO_{50}$-modified surface only in a serum-containing environment (Fig. 4a–f). This result indicated that in a serum-containing environment, $DM_{50}CO_{50}$ promoted cell adhesion by partially following an integrin-dependent mechanism, possibly utilizing RGD motifs of surface-adsorbed proteins; whereas, in a serum-free environment, $DM_{50}CO_{50}$ promoted cell adhesion by following an integrin-independent mechanism, possibly via direct binding to cell surface polysaccharides such as heparin. It is noteworthy that in the presence of EDTA to block integrin-dependent cell adhesion in both serum-free and serum-containing conditions, cell adhesion to $DM_{50}CO_{50}$-modified surface was superior to that on RGD peptide-modified surface, which underpinned our design of cell-adhesive β-amino acid polymers with dual mechanism cell adhesion (Fig. 4g). It is worth mentioning that the RGD motifs of proteins adsorbed on $DM_{50}CO_{50}$-modified surface can interact with cell surface integrins only when these proteins are in an appropriate conformation to present the RGD motifs away from the surface and free to interact with cell surface integrins. Limited by space in Fig. 4g, we only draw the surface-adsorbed serum protein with appropriate conformation to present the RGD motif for integrin binding.

**Effect of adsorbed proteins on cell adhesion and cell growth**. To examine our hypothesis and the above implication of surface-adsorbed proteins in cell adhesion, we analyzed the amount of surface-adsorbed total protein after variable surfaces were incubated with a serum-containing cell culture medium for 2 h when the total protein adsorption already reached saturation (Supplementary Fig. 8). Compared to RGD-modified surface, $DM_{50}CO_{50}$-modified surface adsorbed significantly higher amount of total serum protein (Fig. 5a). Moreover, we characterized the morphology of $DM_{50}CO_{50}$ and RGD-modified surface after serum protein adsorption using AFM (Fig. 5b). We observed more and larger protein aggregates on the $DM_{50}CO_{50}$-modified surface than that on the RGD-modified surface, which may be caused by the ability of the $DM_{50}CO_{50}$-modified surface

to adsorb more serum protein than does the RGD-modified surface. We also did AFM characterization on the surface adsorption of important cell-adhesive single serum protein, such as FN, after surface incubation with the protein for 2 h, and we observed a result similar to total protein adsorption (Fig. 5c). On the $DM_{50}CO_{50}$-modified surface, the abundant and homogeneous distribution of adsorbed protein echoed the evenly adhered cells to the surface. After MC3T3-E1 cells were cultured on the surface for 2 days in a serum-containing environment, quantification on surface-adsorbed individual protein (FN, VN, Coll, and LAM) showed that FN and Coll are the most abundant proteins on $DM_{50}CO_{50}$-modified surface, and the amount of adsorbed FN and Coll on $DM_{50}CO_{50}$-modified surface is also significantly higher than that on the RGD-modified surface (Fig. 5d). According to GO enrichment analysis in RNA-seq analysis, we found that both fibronectin-binding (GO:0001968) and collagen-binding (GO:0005518) terms are in the top three of the molecular function categories, which implies a conclusion same as the above experimental study on cell adhesion mechanism in a serum-containing environment that fibronectin and collagen are two important proteins involved in $DM_{50}CO_{50}$ polymers' function in promoting preosteoblast cell adhesion.

To examine the above hypothesis, we seeded preosteoblast cells on $DM_{50}CO_{50}$-modified surface that was pre-incubated with a serum-containing medium and then with RGD-specific integrin αvβ3 before cell seeding. After 2 h, we observed that αvβ3 attenuated cell adhesion to $DM_{50}CO_{50}$-modified surface by reducing cell spreading area, though without affecting adhered cell density; whereas, αvβ3 substantially attenuated cell adhesion to RGD-modified surface on both spreading area and adhered cell density (Fig. 5e–g). The amount of surface-adsorbed total protein was not affected for both the $DM_{50}CO_{50}$ and RGD-modified surfaces (Supplementary Fig. 9). The substantial reduction of cell spreading area on αvβ3-treated $DM_{50}CO_{50}$ surface indicated that cell-adhesive β-amino acid polymers can exploit adsorbed proteins and expose their RGD motif for direct binding with cell surface integrin; whereas, the unaffected cell density on αvβ3-treated $DM_{50}CO_{50}$ surface indicated that cell-adhesive β-amino acid polymers have additional cell adhesion mechanisms. This result echoed the above EDTA blocking result and both underpinned the dual mechanism cell adhesion of the cell-adhesive β-amino acid polymers (Fig. 5h).

**Polysaccharides-associated cell adhesion**. In order to pinpoint the integrin-independent cell adhesion mechanism, we studied cell adhesion in a serum-free environment to exclude the influence of RGD-containing serum proteins and found that $DM_{50}CO_{50}$-modified surface still enabled excellent cell adhesion, with cell morphology similar to that on surfaces modified with RGD peptide and KRSR peptide (Fig. 6a). This observation implied the integrin-independent cell adhesion mechanism of $DM_{50}CO_{50}$ likely involves direct interactions between the polymer and cell surface membrane proteins or biomolecules. The aforementioned strong protein adsorption property of $DM_{50}CO_{50}$ and our design of KRSR mimicking cell adhesion encouraged us to examine the possible effect of cell surface tethered proteins and cell surface polysaccharides, respectively.

To examine the above hypothesis of RGD-independent cell adhesion, we pretreated preosteoblast cells in a serum-free medium with plasmin, collagenase, heparinase, and hyaluronidase individually to degraded cell surface fibronectin and vitronectin together, collagen, heparin, and hyaluronan, respectively, before cell seeding on variable surfaces and examination of cell density and morphology[30] (Fig. 6a–c). None of these treatments affected cell adhesion on RGD peptide-modified

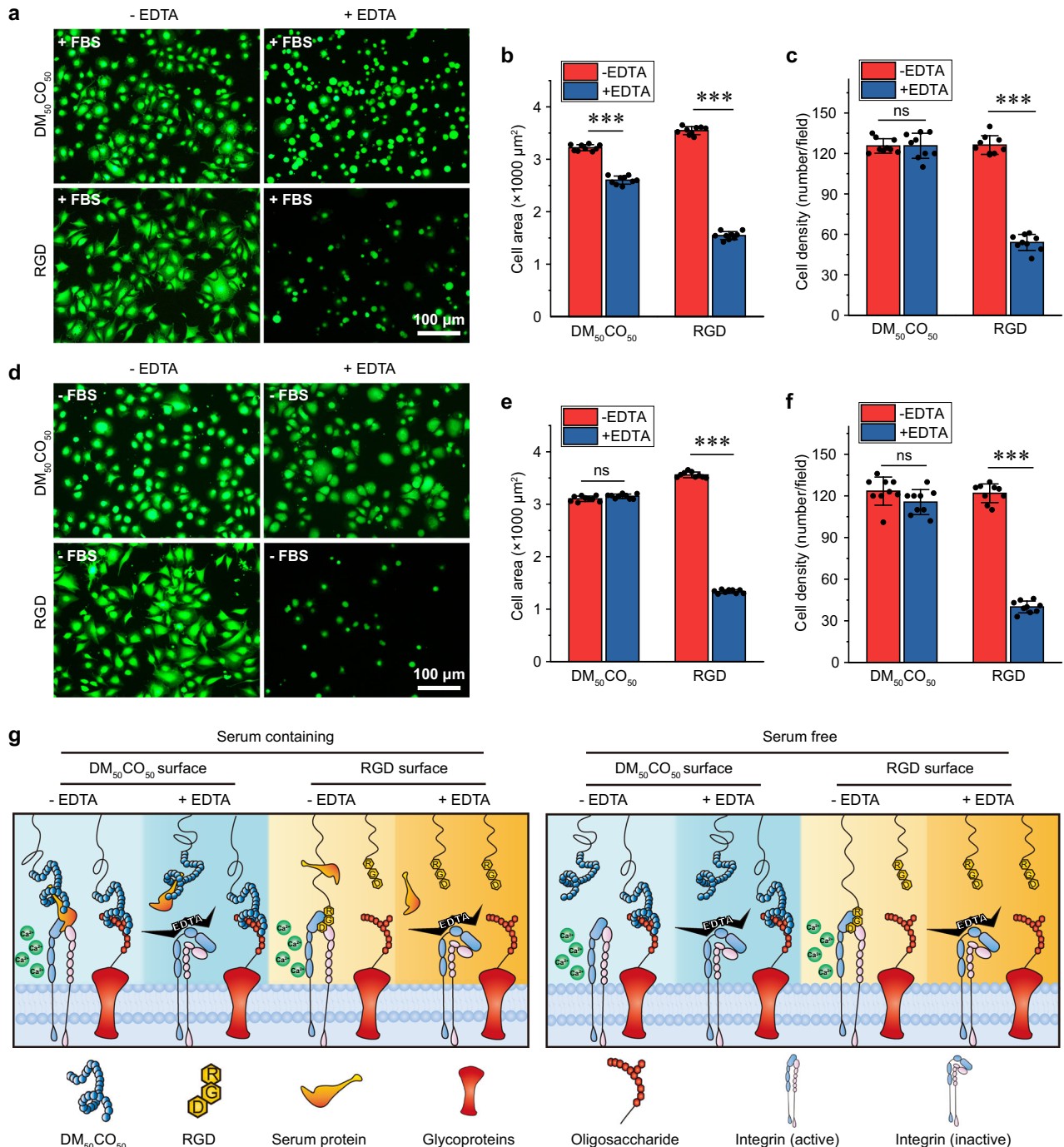

**Fig. 4 Cells adhesion via both integrin-dependent and -independent mechanisms. a** Fluorescent micrographs of live/dead stained cells, **b** cell area analysis, and **c** cell density analysis, of cell adhesion to $DM_{50}CO_{50}$ and RGD-modified surfaces in serum-containing media for 3 h, with or without EDTA treatment. **d** Fluorescent micrographs, **e** cell area analysis, and **f** cell density analysis, of cell adhesion to $DM_{50}CO_{50}$ and RGD-modified surfaces in serum-free media for 3 h, with or without EDTA treatment. **g** Cell adhesion mechanisms of $DM_{50}CO_{50}$ and RGD-modified surfaces in different conditions. Data in (**b**, **c**, **e**, **f**) represent mean ± s.d. ($n = 9$). Statistical analysis: two-tailed $t$-test, ***$p < 0.0001$, ns = not significant.

surface, and only heparinase treatment compromised cell adhesion on KRSR peptide-modified surface, which is consistent to the cell surface integrin-binding and heparin-binding mechanism of RGD peptide and KRSR peptide, respectively[31,32]. In sharp contrast, all enzymatic treatments significantly compromised cell adhesion on $DM_{50}CO_{50}$-modified surfaces, regarding the cell spreading area, which indicated cell surface protein and polysaccharide involved cell adhesion mechanism of $DM_{50}CO_{50}$, in addition to aforementioned integrin-dependent mechanism. In

our RNA-seq analysis as aforementioned in Fig. 3b, we observed glycosaminoglycan-binding (GO: 0005539) term as one of the top molecular function categories in the GO enrichment analysis, which implies a conclusion the same as that in our experimental study on polysaccharide-involved cell adhesion mechanism (Fig. 6). Moreover, both RNA-seq analysis and our experimental study on cell adhesion mechanism echo aforementioned observation on cell morphology (Fig. 2a–c) that polymer $DM_{50}CO_{50}$ supports cell adhesion and spreading different from

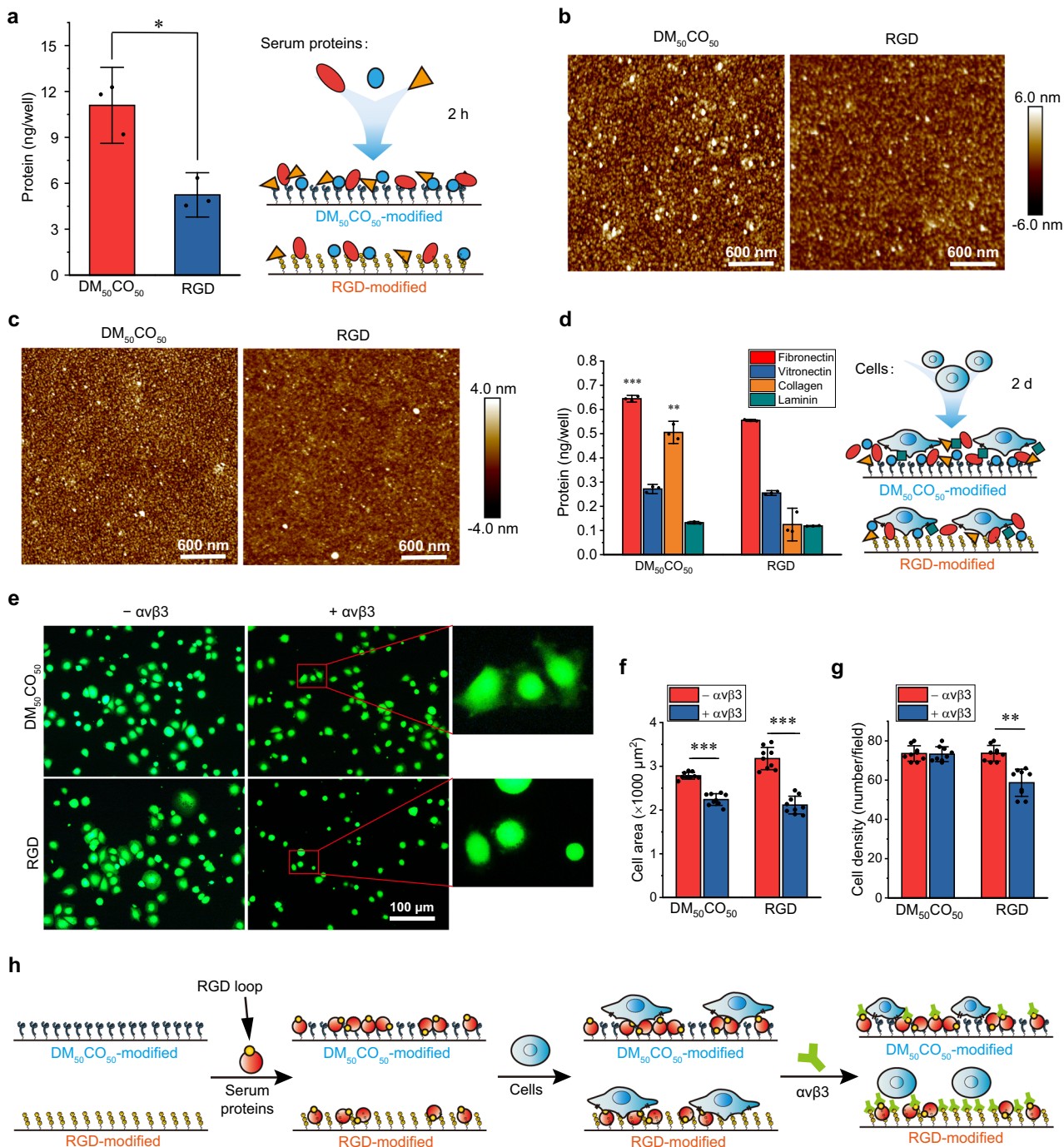

**Fig. 5 Effect of adsorbed serum proteins on preosteoblast cell adhesion to DM$_{50}$CO$_{50}$-modified surfaces. a** The amount of surface-adsorbed total protein after variable surfaces were incubated with a serum-containing cell culture medium for 2 h. **b**, **c** AFM images of DM$_{50}$CO$_{50}$ and RGD-modified surfaces after incubation with serum-containing cell culture medium (**b**) and pure fibronectin (**c**) for 2 h. **d** The amount of surface-adsorbed individual protein (fibronectin, vitronectin, collagen, laminin) on DM$_{50}$CO$_{50}$ and RGD-modified surfaces after cells were cultured on the surfaces for 2 days. Fluorescence micrographs of: **e** live/dead stained cells, **f** spreading area, and **g** density analysis, of cells 2 h after seeding on DM$_{50}$CO$_{50}$ and RGD-modified surfaces, which were first incubated with serum-containing medium and then treated with or without integrin αvβ3 before cell seeding. **h** Carton illustration of results in (**e**–**g**). Data in (**a**, **d**, **f**, **g**) represent mean ± s.d. (*n* = 3 for **a**, **d** and *n* = 9 for **f**, **g**). Statistical analysis: two-tailed *t*-test, **\**p* < 0.001, *\*\**p* < 0.0001.

both RGD and KRSR, as a reflection of cell adhesion mechanism (dual mechanisms) different from either RGD or KRSR.

**Application of cell-adhesive β-amino acid polymer.** It is known that incorporation of cell-adhesive peptides within implants can enhance cell attachment, cell infiltration, and a variety of other

regeneration-favorable response and signaling to promote in vivo tissue regeneration[33,34]. To examine the potential of the optimal cell-adhesive β-amino acid polymer, we did a proof-of-concept demonstration on DM$_{50}$CO$_{50}$-functionalized polyethylene glycol (PEG) hydrogels for bone regeneration using a rat cranial defect model[35]. We chose PEG hydrogel as a hash model for tissue regeneration on purpose, to evaluate the cell adhesion function of

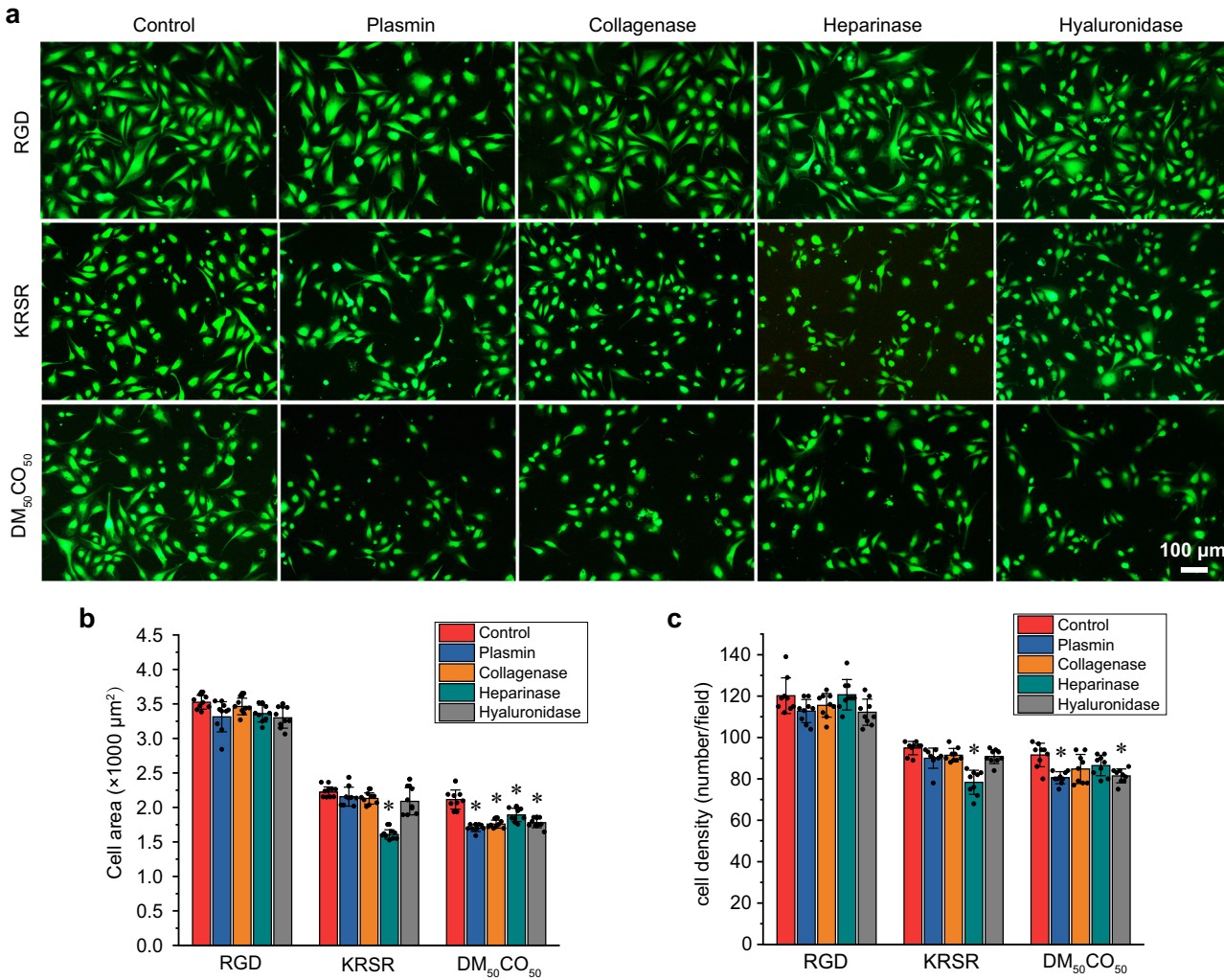

**Fig. 6 Effect of membrane proteins and polysaccharides on preosteoblast cell adhesion.** Fluorescent micrographs of: **a** live/dead stained cells, **b** spreading area, and **c** density analysis, of cells adhered to RGD, KRSR, and $DM_{50}CO_{50}$-modified surfaces for 1 day in a serum-free medium. Cells were treated with plasmin, collagenase, heparinase, and hyaluronidase, respectively, before seeded on the surfaces. Untreated cells were used as the control. Data in (**b**, **c**) represent mean ± s.d. ($n = 9$). Statistical analysis: one-way ANOVA with Tukey post-test, $*p < 0.01$.

$DM_{50}CO_{50}$ and the impact on tissue regeneration, because bare PEG hydrogel has poor performance in tissue regeneration due to its poor cell adhesion and slow degradation. The hydrogel was prepared via crosslinking between an acrylate-terminated 4-arm-PEG (PEG-ACLT, Mw = 10 kDa) and a thiol-terminated 4-arm-PEG (PEG-SH, Mw = 10 kDa) in the presence of a thiol-terminated $DM_{50}CO_{50}$ (HS-$DM_{50}CO_{50}$) at room temperature (rt) (Fig. 7a). The prepared PEG hydrogel from the mold at 3 mm in diameter was swollen in PBS to give the swollen hydrogel for implantation at 5 mm in diameter, in excellent fit to the critical size cranial defect (Fig. 7b). When preosteoblast cells were seeded and cultured for 2 days on top of the hydrogels, cells merely adhered to and clumped on the bare PEG hydrogel; in sharp contrast, $DM_{50}CO_{50}$-functionalized hydrogel strongly promoted preosteoblast cell adhesion and spreading (Fig. 7c). Bare PEG hydrogels, $DM_{50}CO_{50}$-functionalized hydrogels, and two commercial bone repair materials, polylactic acid (PLA)[36] and RGD-containing gelatin methacryloyl (GelMA)[37], were implanted into the cranial defects individually. The result of bone regeneration was examined by micro-CT and Masson's trichrome staining analysis after implantation for 8 weeks (Fig. 7d, e). $DM_{50}CO_{50}$-functionalized PEG hydrogels outperformed the bare PEG

hydrogel group, GelMA group, and PLA group in promoting bone regeneration, as supported by the quantified values of bone volume/total volume ratio (BV/TV) (Fig. 7f). All the above analyses supported the conclusion that incorporation of cell-adhesive β-amino acid polymer to implant scaffold promoted in vivo bone regeneration.

**Discussion**

Fibronectin and RGD peptide are representative cell-adhesive proteins and peptides, respectively, and have been extensively studied to promote cell adhesion. Nevertheless, the prone to proteolysis, difficulty in large-scale preparation, and expensiveness of cell-adhesive proteins and peptides greatly limited their application. Aiming to address these challenges, we designed cationic-hydrophobic amphiphilic β-amino acid polymers as dual mechanism cell-adhesive materials that mimic the cell adhesion mechanisms and functions of RGD and KRSR peptides simultaneously. The optimal polymer strongly promoted preosteoblast cell adhesion by following both an integrin-dependent mechanism to utilize surface-adsorbed proteins bearing RGD motifs and an integrin-independent mechanism to interact directly with cell surface proteins and polysaccharides. The easy synthesis and large

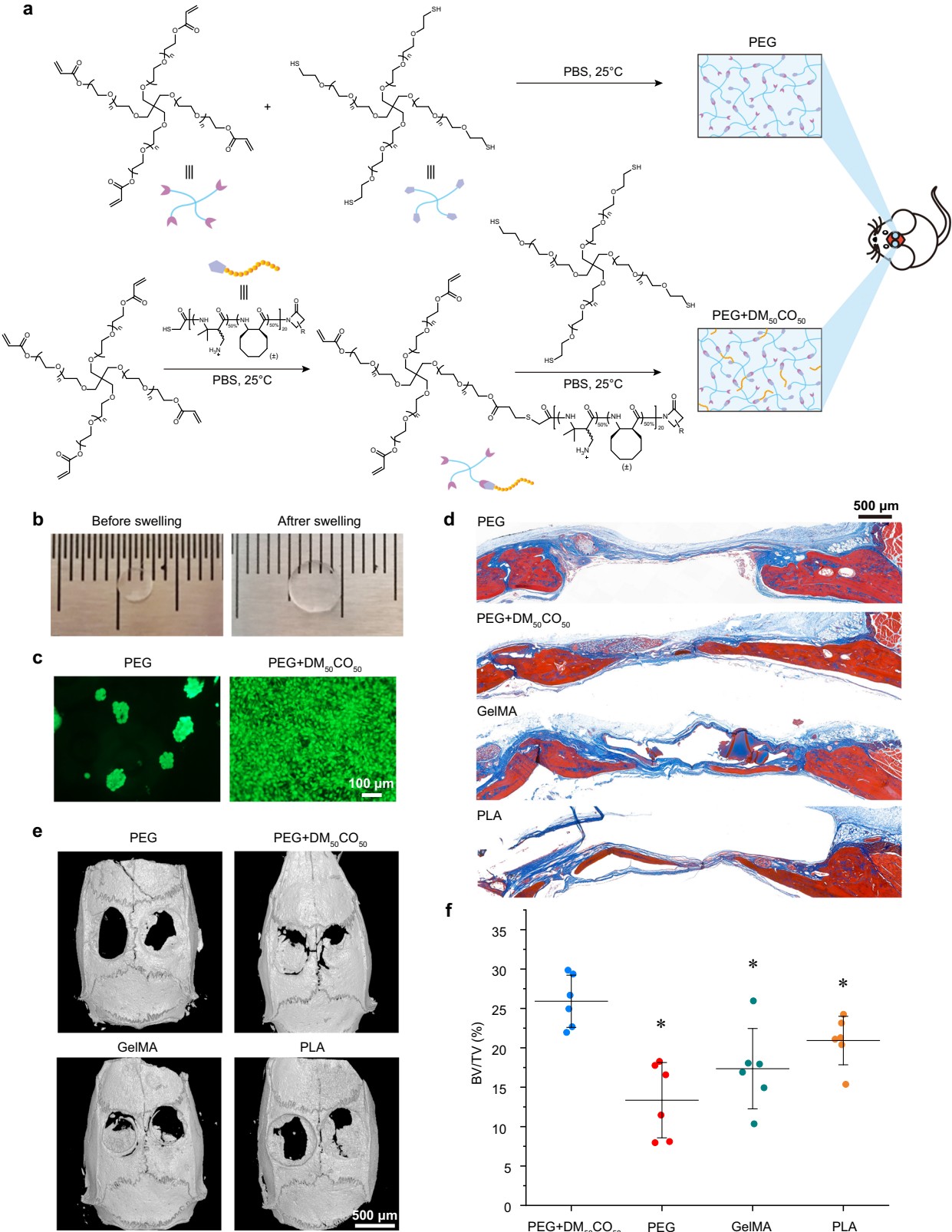

**Fig. 7 Demonstration of cell adhesion impact of DM$_{50}$CO$_{50}$-incorporated hydrogel on in vivo bone regeneration using a critical-size rat calvarial defect model. a** Synthesis of bare PEG hydrogel and DM$_{50}$CO$_{50}$-modified PEG hydrogel. **b** Comparison before and after PEG hydrogel swelling. **c** Preosteoblast cells adhesion to bare PEG hydrogel and DM$_{50}$CO$_{50}$-modified PEG hydrogel after cell seeding for 2 days. Cells were subjected to live/dead staining. **d** Masson's trichrome stain, **e** 3D micro-CT analysis, and **f** bone volume analysis, of the cranial bone samples from bare PEG hydrogels, DM$_{50}$CO$_{50}$-modified PEG hydrogels, GelMA, and PLA after 8 weeks of implantation. Data in (**f**) represent mean ± s.d. ($n = 6$). Statistical analysis: two-tailed $t$-test, *$p < 0.05$.

structural diversity of β-amino acid polymers, as well as the sequence and chirality-independent cell adhesion functions altogether implied great potential of this strategy in designing and developing the next-generation cell-adhesive materials for diverse applications such as cell culture, tissue regeneration, and integration of permanent implants.

## Methods

**Surface preparation**. Surfaces modified with peptides and polymers were prepared using our recently developed method[27]. Glass slides (25 × 76 mm) were cleaned by a UV-ozone cleaner for 25 min, followed by immersion in a solution of 2% (v/v) APTES in anhydrous toluene at rt overnight. After washing steps, APTES-modified glass slides were annealed at 80 °C for 3 h. Then the glass slides were covered with 50-well or 8-well coverslips, followed by adding 10 or 30 μL of the Maleimide-NH-PEG8-CH₂CH₂COONHS ester (OEG8) solution (20 mM in PBS, pH 7.4) into each individual well on the glass slides and incubating the slides at rt for 2 h to obtain the maleimide-functionalized OEG8 surfaces. For peptide and polymer modification, the OEG8 surface was incubated with RGDSPC, KRSRGYC, or β-amino acid polymer solution (0.6 mM in pH 7.4 PBS containing 10% glycerol) in each individual well on the glass slides at rt overnight, and then a solution of thioglycerol at 100 mM in PBS was added to the surface and the slides were incubated at rt for 2 h. Finally, these slides were cleaned by Milli-Q water and dried with N₂ for further use.

**Surface characterization**. XPS was conducted on a Thermo Fisher Scientific ESCALAB 250Xi XPS System equipped with a monochromatic AlKRsource (KE = 1486.6 eV), with the vacuum of analysis room at $8 \times 10^{-10}$ Pa. The bare glass, NH₂-glass, OEG8, and DM₅₀CO₅₀-modified surfaces were analyzed.

The water contact angles of bare glass, OEG8, RGD, and DM₅₀CO₅₀-modified surfaces were measured with a contact angle meter (JC2000D2) from Shanghai Zhongchen Digital Technology Apparatus Co., Ltd. Liquid drops (Millipore water) were delivered to the surface using a microsyringe.

The morphologies of the bare glass, RGD, and DM₅₀CO₅₀-modified surfaces were obtained by AFM with Nanoscope VIII a microscope system (Bruker, USA) in PeakForce Tapping mode. Moreover, the AFM characterization was performed on RGD and DM₅₀CO₅₀-modified surfaces, after incubating these surfaces for 2 h with a fibronectin solution (0.5 μg/mL) or α-MEM medium containing 10% FBS.

**Cell attachment analysis**. A mouse calvarial preosteoblast cell line MC3T3-E1 was cultured in α-MEM medium (containing 10% FBS, 100 U/mL penicillin, 100 μg/mL streptomycin, and 2 mM L-glutamine) at 37 °C in a 5% CO₂ humidified atmosphere. Cells seeding on surfaces modified with variable peptide and polymer were detached with 0.25% trypsin (Gibco) containing 0.02% EDTA, then centrifuged and resuspended in culture medium to a final concentration of $1 \times 10^5$ cells/mL. An aliquot of 10 μl this cell suspension was added to each individual well of the peptide and polymer-modified glass slide to have a seeding density of 140 cells/mm². The slide was placed in a petri dish and incubated for 2 h at 37 °C for cell attachment, followed by addition of fresh medium into the petri dish to immerse the entire slide. After culture for 2 days, the slide was stained with a live/dead staining solution containing 2 μM calcein AM and 4 μM ethidium homodimer-1 for 15 min. Cell adhesion was quantified by scanning the whole slide with ImageQuant LAS 4000 (GE Healthcare). The data were analyzed using ImageQuant LAS 4000 Control software.

**Immunofluorescence analysis of cell morphology**. The degree of cell spreading on a functionalized glass slide was measured by immunofluorescence analysis. After culture medium was removed, the slide was washed with PBS and then cells on the slide were fixed with 4% paraformaldehyde for 20 min. The slide was washed with PBS thrice and incubated for 5 min with 0.4% triton-X in PBS. Subsequently, the slide was washed with PBS thrice, followed by blocking with 3% bovine serum albumin (BSA) in PBS for 1 h at rt and then incubated with Alexa Fluor-555-conjugated anti-vinculin antibody (1:300 in PBS) overnight at 4 °C. The slide was washed with PBS thrice and incubated for 2 h with FITC-phalloidin (1:200 in PBS) and 4′-6-diamidino-2-phenylindole (DAPI) (5 μg/mL) for 10 min. The slide was washed with PBS thrice and its fluorescent images were collected on a confocal microscope (Nikon A1R, Japan).

**Cell proliferation assay**. The Alamar Blue assay was employed to quantitatively determine the proliferation of MC3T3-E1 cells on different surfaces. Cell suspension (30 μL/well) with a density of $4 \times 10^4$ cells was added to each well that was formed by an 8-well silicon coverslip on a glass slide. On days 1, 3, and 5, medium was removed from each well followed by addition of 50 μL/well Alamar Blue (10%, v/v) into each well and incubation for 3 h at 37 °C. The solution in each well was then transferred to a 384-well plate and the fluorescence intensity was quantified using a microplate reader ($\lambda_{ex}$ = 560 nm; $\lambda_{em}$ = 590 nm).

**RNA-seq and data analysis**. MC3T3-E1 cells were cultured for 2 days on DM₅₀CO₅₀ or RGD-immobilized 6 cm diameter glass plates. Total RNA from MC3T3-E1 was extracted with RNA extraction reagent (Servicebio). RNA sample was qualified and quantified using a Nano Drop and Agilent 2100 bioanalyzer (Thermo Fisher Scientific, MA, USA). The RNA sequencing libraries were constructed and sequenced on the BGISEQ-500 platforms.

**The inhibition assay of EDTA**. The preosteoblast cells were treated with 5 mM EDTA at 37 °C for 10 min before cell seeding in serum-free or serum-containing environment. After treatment, the cell adhesion assay was performed as described above for 3 h. Cells were stained with a live/dead staining solution, and then the fluorescent images were captured under a microscope (Nikon Eclipse Ti-S, Japan) at eight different locations of each well. The collected images were analyzed using the Image-pro plus software.

**Quantification of protein adsorption to polymer-modified surfaces**. To quantify initial total protein adsorption onto the DM₅₀CO₅₀ and RGD-modified surfaces from serum-containing culture medium in the absence of cells, an aliquot of 10 μL α-MEM medium containing 10% FBS was added to each well (formed by a 50-well silicon coverslip on top of a glass slide) and the slide was incubated in a humidified atmosphere at 37 °C for 2 h. The slide was washed thrice with PBS and dried with N₂. The amount of total protein adsorbed to surfaces was analyzed using the NanoOrange assay. In brief, an aliquot of 10 μL 1X NanoOrange working solution was added to each well and the slide was incubated for 1 h at rt, followed by scanning the slide with ImageQuant LAS 4000. Data process was conducted using a generated BSA standard curve.

In order to quantify the amount of fibronectin (FN), vitronectin (VN), collagen (Coll), and laminin (LAM) on MC3T3-E1 cultured surface for 2 days, we used a method that combined NanoOrange assay and immunodetection. Taking the FN quantification for example, using slides that were covered by 50-well silicon coverslips, a series of 10 μL of FN solutions (with concentration range from 0.05 to 51.2 μg/mL) were added to each well on two slides (slide 1 and slide 2). These two slides were incubated in a humidified atmosphere at 37 °C for 2 h and then were washed with PBS to removed non-adsorbed protein to provide both slides with the same amount of surface adsorbed FN at a specific initial FN incubating concentration. Slide 1 was analyzed using NanoOrange reagent. Slide 2 was incubated with rabbit anti-FN antibody (1:400 dilution) and FITC-conjugated goat anti-rabbit IgG secondary antibody (1:400 dilution) sequentially to obtain the fluorescence values corresponding to different initial FN incubating concentration. Meanwhile, a FN standard curve was generated on slide 3 using mixtures of FN solution and NanoOrange reagent, and this standard curve was used in combination with the NanoOrange data from slide 1 to quantify the actual surface adsorbed FN in both slide 1 and slide 2. Then, from slide 2, we can generate a calibration curve between the fluorescence value from the secondary antibody and the quantity of surface adsorbed FN. From this calibration curve, we can quantify the surface adsorbed FN from actual sample after cell culture for 2 days using rabbit anti-FN antibody (1:400 dilution) and FITC-conjugated goat anti-rabbit IgG secondary antibody (1:400 dilution). Other individual proteins (VN, Coll, LAM) were quantified using the same protocol, using rabbit anti-VN (1:400 dilution), rabbit anti-collagen I (1:320 dilution), rabbit anti-LAM (1:300 dilution), and FITC-conjugated goat anti-rabbit IgG secondary antibody (1:400 dilution).

**Integrin αvβ3 to block RGD**. An aliquot of 10 μL α-MEM medium containing 10% FBS was added to each well (formed by a 50-well coverslip on the top of a glass slide) and the slide was incubated in a humidified atmosphere at 37 °C for 2 h[24]. The slide was washed with PBS and incubated with integrin αvβ3 (10 μg/mL in PBS) at 37 °C for 1 h. Then cells at a concentration of $1.5 \times 10^5$ cells/mL in a serum-free medium were added to each individual well, having bottom surface modified with DM₅₀CO₅₀ or RGD peptide. After the slide was incubated at 37 °C for 2 h, cells in each well were stained with a live/dead staining solution and the fluorescent images were captured under a microscope (Nikon Eclipse Ti-S, Japan) at eight different locations in each well. The collected images were analyzed using the Image-pro plus software.

**Cell adhesion in a serum-free environment**. Cells were detached and washed thrice with a serum-free medium. In order to examine the impact of cell membrane-tethered proteins on cell adhesion in a serum-free environment, cells were suspended and incubated at 37 °C for 20 min in 3 mL serum-free α-MEM medium that contains 0.307 U/mL plasmin, 10 U/mL collagenase (types I and II), 2 U/mL heparinase (types I, II, and III), or 10 U/mL hyaluronidase, respectively. The protease-treated cells were washed and suspended in a serum-free α-MEM to a final concentration of $1.5 \times 10^5$ cells/mL. An aliquot of 10 μL this cell suspension was added to each individual well (formed by a 50-well silicon coverslip on the top of a glass slide) that has bottom surface modified with DM₅₀CO₅₀, RGD, or KRSR, respectively, and then the slide was placed in a petri dish for incubation at 37 °C for 2 h for cell attachment. Then additional fresh medium was added into the petri dish to immerse the entire slide and the whole dish was incubated for 1 day. After removing the medium, cells were stained with a live/dead staining solution and the fluorescent images were captured under a microscope (Nikon Eclipse Ti-S, Japan)

at different locations of each well. The collected images were analyzed using Image-pro plus software.

**PEG hydrogel preparation**. Thiol-ended $DM_{50}CO_{50}$ was mixed at a 1:10 molar ratio with the acrylate-terminated 4-arm-PEG (PEG-ACLT, Mw = 10 kDa) to a final concentration of 5 wt% in PBS, and the mixture was shaken for 1 h. Then thiol-terminated 4-arm-PEG (PEG-SH, Mw = 10 kDa, dissolved in PBS at 5 wt%), with the same equivalent of 4-arm-PEG-ACLT, was added to the mixture followed by blending immediately and quick transfer of 7 μL the mixture to each well in a silicone mold with 3 mm diameter to form hydrogels within minutes. After 1 h, hydrogels were transferred by a tweezer to PBS that was replaced 5 times with an interval of 12 h, then we obtained the hydrogel at a final diameter at 5 mm, which matches the critical size cranial defect. Hydrogels without incorporation of β-amino acid polymer were used as the control.

**Gelatin methacryloyl (GelMA) hydrogel preparation**. GelMA was dissolved in PBS at a concentration of 10% (w/v) with 0.25% (w/v) lithium phenyl-2,4,6-trimethylbenzoylphosphinate (LAP). The mixture was heated to 40 °C for 30 min until completely dissolved. An aliquot of 17 μL this mixture was added into each well of a silicone mold with 4.5 mm diameter and crosslinked to form hydrogels by blue light irradiation (wavelength of 405 nm). Before implantation, these hydrogels were swollen in Milli-Q water to reach the final diameter at 5 mm.

**Polylactic acid (PLA) disc preparation**. PLA, 10% (w/v), was dissolved in dichloromethane. The solution was poured onto a glass plate and allowed to naturally volatilize for 24 h at rt to obtain a large PLA sheet, which was punched into discs (5 mm in diameter) for implantation directly.

**Rat cranial defect model for in vivo bone regeneration study**. All animal procedures were done in accordance with the Guidelines for Care and Use of Laboratory Animals of the Ninth People's Hospital, Shanghai Jiao Tong University School of Medicine and experiments were approved by the Animal Ethics Committee of the Ninth People's Hospital, Shanghai Jiao Tong University School of Medicine. Female Sprague-Dawley rats (SD rats, 8 weeks old) were anesthetized using 50 mg/kg pentobarbital sodium. Then a 3 cm long, sagittal incision was made on the skin and down to the cranium. After peeling the periosteum, two parallel 5 mm critical cranial defects were obtained by drilling into each rat using a 5 mm diameter trephine. After samples were implanted into the defects, the incision was closed. The rats were fed up to 8 weeks for bone regeneration evaluation.

**X-ray computed tomography (Micro-CT) analysis**. Animals were sacrificed after 8 weeks post-operation, and the skulls were collected for fixation with 4% paraformaldehyde (n = 6). The bone tissue in the defect area was scanned with micro-CT (SkyScan 1172, SkyScan, Aartselaar, Belgium). The bone volume (BV)/tissue volume (TV) was calculated using CT Analyser software.

**Histological analysis**. After Micro-CT evaluation, tissue samples were subjected to histological analysis. The skulls were decalcified in 10% EDTA, dehydrated in a graded ethanol series and embedded in paraffin wax. Sections of each sample at 5 μm thickness were cut and mounted onto slides for Masson's trichrome staining. Then these slides were scanned under bright field using a scanner (Pannoramic 250/MIDI) equipped with the CaseViewer 2.0 software.

**Statistics and reproducibility**. Statistical analysis was performed with Origin software. Significance between two groups was determined by $t$-test. One-way analysis of variance (ANOVA) with Tukey post-test for more than two variables was carried out. All results were expressed as mean ± standard error. All micrograph assays were carried out at least three independent times with similar results.

**Reporting summary**. Further information on research design is available in the Nature Research Reporting Summary linked to this article.

## Data availability
Data that support the findings detailed in this study are available in the supplementary information and this article. RNA-seq data have been deposited in the NCBI Gene Expression Omnibus database under accession code PRJNA669614. Any other source data perceived as pertinent are available, on reasonable request, from the corresponding author. Source data are provided with this paper.

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

## Acknowledgements

We thank Dr. Yibing Wang and Qize Xuan at East China University of Science and Technology for their help on surface characterization using AFM. This research was supported by the National Natural Science Foundation of China for Innovative Research Groups (No. 51621002), the National Natural Science Foundation of China (No. 21774031, 31800801), the National Key Research and Development Program of China (2016YFC1100401), the Natural Science Foundation of Shanghai (18ZR1410300), Program of Shanghai Academic/Technology Research Leader (20XD1421400), Research program of State Key Laboratory of Bioreactor Engineering, and the Fundamental Research Funds for the Central Universities (22221818014). We thank the beamline BL13W of the Shanghai Synchrotron Radiation Facility (SSRF, Shanghai, People's Republic of China) for the help with the micro-CT analysis. We also thank the Research Center of Analysis and Test of East China University of Science and Technology for the help with the characterization.

## Author contributions

R.L. directed the whole project. Q.C. and R.L. conceived the idea, proposed the strategy, designed the experiments, evaluated the data, and wrote the manuscript together. Q.C. performed majority of the experiments. D.Z. performed the surface characterization and hydrogel preparation, participated in chemical synthesis, RNA-seq, and Micro-CT analysis. W.Z. and Y.Y. participated in the Micro-CT and histological analysis. H.Z. participated in chemical synthesis. J.Z. drew the 3D schematic diagram. M.C. and J.L. participated in the in vivo experiments. All authors discussed the results and commented on the manuscript.

## Competing interests

R.L. and Q.C. are co-inventors on a patent covering the function of β-amino acid polymers presented in this report. The remaining authors declare no competing interests.
