## [Peer Review File · Nature Communications]

REVIEWER COMMENTS

Reviewer #1 (Remarks to the Author):

This manuscript reports on preosteoblast adhesion to surfaces modified with synthetic amphiphilic β -amino acid polymers. These are compared to surfaces modified with the RGD and KRSR peptides, well known to promote cell adhesion via integrin receptors on cell surfaces. They are believed to be superior to the conventional peptides in that they are resistant to proteolysis and can readily be scaled up. The work is of high quality, comprehensive, and carefully carried out. It makes a significant scientific contribution, and may also contribute to materials development for tissue engineering and other materials-related biomedical applications. Some concerns:

1. A major conclusion is that the synthetic polymers interact by two mechanisms, both an integrin-dependent mode and a mode involving binding of the cationic residues to anionic cell surface polysaccharides, while the RGD interacts only via cell surface integrins. This is based largely on experiments in the presence of EDTA which show that adhesion remains significant on the β -amino acid polymer whereas it is greatly reduced on the RGD. Experiments with $\alpha v\beta 3$ antibody against the integrin also provide some support for a dual mechanism as do experiments with cells treated with various enzymes.

One question in all of this is the mechanism by which the β -amino acid polymer binds specifically to fibronectin/vitronectin in the presence of serum, presumably leaving the RGD motif free to interact with the integrin as shown in Fig 4g? Can the authors elaborate on this?

2. The experiments on the role of adsorbed proteins need clarification and more explanation/discussion. The data (Fig 5) are based on immunoassay and it is not clear if the fluorescence intensities reflect the relative quantities of the 3 (or 4) different proteins? Do the relative quantities change over time? Are the surfaces saturated (i.e. maximally covered) in the 2-hour time frame? Can the data be calibrated to get absolute values?

3. It appears that adsorption increases when cells are cultured on the protein layers (Fig 5a vs 5b). How is this explained? Is there additional adsorption from the cell culture medium indicating that saturation was not achieved in the initial exposure to serum (Fig a)? If so what is the significance?

4. In the experiments where serum proteins are adsorbed, then treated with $\alpha v\beta 3$ antibody, then cultured with cells, what are the quantities of proteins adsorbed after cell culture?

5. What about the many other, more abundant proteins in serum? I would assume they are equally likely to adsorb to these polymers via hydrophobic or ionic interactions.

6. The surface characterization data are important and at least a synopsis of these should be brought into the main body of the paper and some discussion given. More discussion of the cell adhesion and protein adsorption results should be given in relation to these data.

7. The acronyms for the subunits: CP, CH, CO, NM, MM, DM, HLys should be explained.

8. The English is generally very good but a few "Sinocisms" persist and should be corrected, especially the frequent absence (or incorrect presence) of the plural "s".

Reviewer #2 (Remarks to the Author):

In this manuscript Chen Q. et al present an interesting work on the use of cationic-hydrophobic amphiphilic β -amino acid polymers to generate new cell adhesive materials. The work presented here may have the potential to develop novel cell adhesive materials that resist to protease, cheap and easy to prepare in large quantity. The in vitro results obtained on pre-osteoblastic cells are convincing and the β -amino acid polymers give significantly higher adhesion by comparison to classically used adhesive peptides such as RGD and KRSR. However, some experiments are wrongly described and more importantly, some experiments are insufficiently interpreted. In addition, the in vivo results are less convincing than in vitro results. More widely, the paper is missing a correct discussion of the presented results.

1) From their Figure 2 the authors assert that their DM50CO50 polymer implies a different cell

adhesion mechanism from that of RGD and KRSR peptides. It appears rather that the adhesion is similar to RGD peptide but delayed in time. The mechanism cannot be concluded from this study. Moreover, the authors did not discuss the cell spreading which appears lower on DM50CO50 than on RGD peptide notably after 24h. However, for concluding if there is a real difference in term of spreading and if it's not an effect of cell density, the area of individual cells must be quantified and compared between the surfaces at 3 and 24 hours.

2) The RNA-seq analysis is really interesting and give important results about the difference between adhesive peptides and DM50CO50 polymer's influence on cells. The most differences concern the downstream effect of ECM reorganization as highlighted by the authors. However, other GO terms are not discussed by the authors while their -lgP is relatively high (Fig 3b). In particular, several GO terms relative to cell motility or cell division appear in high position in biological processes compared to ECM reorganization and must be also discussed. Indeed, after influencing their adhesion, the DM50CO50 polymer could also influence strongly the migration capacity of preosteoblastic cells and their capacity to colonize new areas during bone regeneration. In addition, the same can be seen in the KEGG pathway enrichment results with the highest score for the ECM-interaction pathway. However, the Wnt signalling pathway like focal adhesion pathway are also present with a relatively high -lgP (Fig. 3c). These two pathways have been shown to be involved in cell migration and cell proliferation. In conclusion, a deeper discussion of this RNA-seq analysis is missing in this paper.

3) In the further parts, the authors have smartly demonstrated that in a serum-containing environment, DM50CO50 promoted cell adhesion by partially following an integrin-dependent mechanism whereas, in a serum-free environment, DM50CO50 promoted cell adhesion by following an integrin-independent mechanism, possibly via direct binding to cell surface polysaccharides such as heparin. These results were confirmed using an anti $\alpha\beta3$ integrin antibody or by degrading cell surface polysaccharides. Therefore, it could be interesting to discuss these results with the RNA-seq results and the different highlighted signal transduction pathways.

4) An anti $\alpha\beta3$ antibody was used to block the interaction of the $\alpha\beta3$ integrin with the RGD peptide in cells adhering on surfaces modified with RGD peptide or the DM50CO50 polymer. However, the description of this experiment is unclear as well in the results part as in the materials and method part showing that this experience was not well understood by the writer, casting doubt on its validity. Why the antibody against $\alpha\beta3$ integrin is incubated with the modified surfaces before seeding the cells? What will the antibody interact with? Actually, the correct experience would be to treat the cells with the anti $\alpha\beta3$ integrin antibody before to seed them on the different surfaces. In the materials and method part, a better title would be "Antibody anti $\alpha\beta3$ integrin to block cell adhesion through RGD" instead of "Antibody $\alpha\beta3$ to block RGD". The authors must rewrite the materials and method as well as the results parts to make this experiment clearer and convince the reader of its validity.

5) The in vivo demonstration of the efficiency of DM50CO50 to promote bone formation is not really convincing since the bone formation increases from 5% with the control material (PEG) to 10% of the total volume of the cavity which is still 90% empty. Therefore, the authors' assertion that their scaffold greatly promotes in vivo bone regeneration is really exaggerated and needs to be revised.

Minor points:

- The preosteoblastic cells used by the authors are referred in the Materials and methods as MC3T3 cells. The sub-line must be given. Classically, it's the sub-line E1 that is used.
- Fig. 5 caption: the authors should choose between protein density or protein adsorption. A density is generally expressed in function of the area. Can the technique used really quantify the protein density?
- Materials and methods part on hydrogel preparation: the authors describe a silicon mould (and not mode) with 3 mm in diameter while the critical size cranial defect was drilled using a 5 mm

diameter trephine. What is the correct diameter size for the hydrogel used in the in vivo experiment?

- Materials and methods part on histological analysis: "After microangiography analysis, tissue samples were subjected to histological analysis". What is this microangiography analysis?

Reviewer #1 (Remarks to the Author):

This manuscript reports on preosteoblast adhesion to surfaces modified with synthetic amphiphilic β -amino acid polymers. These are compared to surfaces modified with the RGD and KRSR peptides, well known to promote cell adhesion via integrin receptors on cell surfaces. They are believed to be superior to the conventional peptides in that they are resistant to proteolysis and can readily be scaled up. The work is of high quality, comprehensive, and carefully carried out. It makes a significant scientific contribution, and may also contribute to materials development for tissue engineering and other materials-related biomedical applications. Some concerns:

Response: We thank the reviewer for the favorable comments and specific questions below.

1. A major conclusion is that the synthetic polymers interact by two mechanisms, both an integrin-dependent mode and a mode involving binding of the cationic residues to anionic cell surface polysaccharides, while the RGD interacts only via cell surface integrins. This is based largely on experiments in the presence of EDTA which show that adhesion remains significant on the β -amino acid polymer whereas it is greatly reduced on the RGD. Experiments with $\alpha\beta3$ antibody against the integrin also provide some support for a dual mechanism as do experiments with cells treated with various enzymes.

One question in all of this is the mechanism by which the β -amino acid polymer binds specifically to fibronectin/vitronectin in the presence of serum, presumably leaving the RGD motif free to interact with the integrin as shown in Fig 4g? Can the authors elaborate on this?

Response: We thank the reviewer for the question that reminds us to provide extra explanation and clarify on this important point. The EDTA-resulted integrin blocking study indicated that in a serum-containing environment, DM₅₀CO₅₀ promoted cell adhesion by partially following an integrin-dependent mechanism, possibly utilizing RGD motifs of surface adsorbed proteins. Serum proteins such as fibronectin, vitronectin and collagen can be adsorbed on the surface of DM₅₀CO₅₀ at variable conformation. When these serum proteins are on the DM₅₀CO₅₀ surface in an appropriate conformation to present the RGD motifs away from the surface, these RGD motifs are free to interact with the integrin on the cell surface as shown in Figure 4g. In Figure 5h, a different experiment depicts that the integrin $\alpha\beta3$ blocks the RGD motifs of surface adsorbed serum proteins, which indicated that RGD motifs in surface adsorbed proteins do participate cell adhesion to DM₅₀CO₅₀

surface. We proposed that surface adsorbed serum proteins can have variable conformation on the DM₅₀CO₅₀ surface, and adsorbed proteins with appropriate conformation can present their RGD motifs away from the surface and free to interact with cell surface integrin. Limited by space in figure 4g, we only drew the adsorbed serum protein with appropriate conformation to present the RGD motif for integrin binding. Related discussion was added to our revised manuscript.

Fig. 4 | g. Cell adhesion mechanisms of DM₅₀CO₅₀ and RGD modified surfaces in different conditions.

Fig. 5h | h. Cartoon illustration of results in 5e-g.

2. The experiments on the role of adsorbed proteins need clarification and more explanation/discussion. The data (Fig 5) are based on immunoassay and it is not clear if the fluorescence intensities reflect the relative quantities of the 3 (or 4) different proteins? Do the relative quantities change over time? Are the surfaces saturated (i.e. maximally covered) in the 2-hour time frame? Can the data be calibrated to get absolute values?

Response: We thank the reviewer for this comment. Fluorescence intensity can reflect the relative quantity of each individual protein. Due to the unpredictable conformation of surface adsorbed proteins, it is hard to have the ratio of 1st antibody to each individual protein on the surface, which implies that fluorescence intensity is hard to reflect the relative quantity between different proteins. We have followed suggestion from the reviewer to quantify the surface adsorbed proteins (FN, VN, Coll and LAM) on MC3T3-E1 cultured surface after 2 days incubation, using a method that combined NanoOrange assay and immunodetection. This quantification analysis showed that the adsorbed protein on DM₅₀CO₅₀ modified surface is 0.64 ng/well, 0.27 ng/well, 0.5 ng/well, and 0.1 ng/well for FN, VN, Coll and LAM respectively, as shown in the figure below (Figure 5d in our revised manuscript).

Fig. 5 | d. The amount of surface adsorbed individual protein (fibronectin, vitronectin, collagen, laminin) on DM₅₀CO₅₀ and RGD modified surfaces after cells were cultured on the surfaces for 2 days.

For surfaces incubated with 10% FBS containing medium, we tried NanoOrange kit to quantify the amount of each individual protein (FN, VN, Coll), which, however, is below the detection limit of NanoOrange assay. We then also tried the very sensitive surface plasmon resonance (SPR) to quantify the amount of each individual protein (FN, VN, Coll) using corresponding antibodies and immunoassay involving a BSA blocking step to alleviate nonspecific binding of antibody to the surface adsorbed proteins. However, such complicated operation to detection a specific protein, such as FN, within a whole population of surface adsorbed serum proteins, comparing to normal SPR detection on direct binding of protein adsorption, leads to a baseline fluctuation that hampered our analysis.

Nevertheless, we are able to quantify the total protein adsorbed on the surface to be 11.1 ng/well and 5.2 ng/well for DM₅₀CO₅₀ and RGD modified surface using NanoOrange assay, as shown in the figure below (Figure 5a in our revised manuscript). As our discussion below in question 5 from the reviewer, we agree with the reviewer that in addition to the three proteins (FN, VN and COLL), other proteins, such as BSA in the serum, can also be adsorbed to the material surface. The quantification of total protein adsorption can better reflect the real protein adsorption capacity between the DM₅₀CO₅₀ and RGD modified surface.

Fig. 5 | a. The amount of surface adsorbed total protein after variable surfaces were incubated with a serum-containing cell culture medium for 2 hours.

To address the reviewer's question if the surfaces saturated in the 2-hour time frame, we did extra experiment. We used NanoOrange assay to detect the total protein adsorption from serum-containing medium at the three time points, 1 hour, 2 hours and 3 hours. We found that protein adsorption reached saturation at 1 hour already as shown below (Supplementary Fig. 7 in our revised manuscript). This result is consistent to the conclusion in precedent literatures that usually within 1-2 h protein adsorption can reach saturation. Therefore, variable incubation time (is 0.5 h, 1 h, 1.5 h) for testing the amount of protein adsorption on the surface have been used in precedent literature (*ACS Appl. Mater. Interfaces* **2018**, 10, 8, 6879–6886; *Adv. Mater.* **2008**, 20, 335–338; *Adv. Mater.* **2017**, 1700617).

Supplementary Figure 7. Quantification of adsorbed serum protein on DM₅₀CO₅₀ and RGD modified surfaces after incubation with serum-containing cell culture media for 1, 2 and 3 hours.

We have updated all above discussion into our revised manuscript that “we analyzed the amount of surface adsorbed total protein after variable surfaces were incubated with a serum-containing cell culture medium for 2 hours when the total protein adsorption already reached saturation (Supplementary Fig. 7). Compared to RGD modified surface, DM₅₀CO₅₀ modified surface adsorbed significantly higher amount of total serum protein (Fig. 5a) On the DM₅₀CO₅₀ modified surface, the abundant and homogeneous distribution of adsorbed protein echoed the evenly adhered cells to the surface. After MC3T3-E1 cells were cultured on the surface for 2 days in a serum-containing environment, quantification on surface adsorbed individual protein (FN, VN, Coll and LAM) showed that FN and Coll are the most abundant proteins on DM₅₀CO₅₀ modified surface, and the amount of adsorbed FN and Coll on DM₅₀CO₅₀ modified surface is also significantly higher than that on the RGD modified surface (Figure 5d).”

3. *It appears that adsorption increases when cells are cultured on the protein layers (Fig 5a vs 5b). How is this explained? Is there additional adsorption from the cell culture medium indicating that saturation was not achieved in the initial exposure to serum (Fig a)? If so what is the significance?*

Response: Although the protein adsorption has already reached saturation on the surface after incubation with a serum-containing cell culture medium for two hours as discussed above to answer question 2 from the reviewer, during the 2 days culture on the surface cells will undergo a series of physiological activities such as cell proliferation and migration. All of these cell behaviors are related to extracellular matrix (ECM) proteins, such as those four proteins in our study. Cells can digest and secrete ECM proteins according to their own need and thus adjust the protein on the substrate surface. As our discussion in above question 2 from the reviewer, we agree with the reviewer that fluorescence intensity can reflect the relative quantities of each individual protein but hard to reflect the relative quantities between different proteins. Therefore, we quantified the amount of surface adsorbed protein and substituted the original Figure 5a and 5b (fluorescence intensity) with new Figure 5a and 5d (quantified amount of protein) in our revised manuscript.

4. *In the experiments where serum proteins are adsorbed, then treated with αvβ3 antibody, then cultured with cells, what are the quantities of proteins adsorbed after cell culture?*

Response: To answer this interesting question, we quantified protein adsorption after cell culture

on a serum protein adsorbed surface with or without integrin $\alpha\beta3$ -blocking. We found no difference in the quantities of proteins adsorbed after cell culture on either surface as shown in the figure below (Supplementary Fig. 8 of our manuscript); the quantity of adsorbed protein on DM₅₀CO₅₀ modified surface, with or without integrin $\alpha\beta3$ blocking, is 9.6 ng/well and 10.4 ng/well, respectively. This result is comparable to the protein adsorption amount (11.1 ng/well) on the surface incubated with a serum-containing cell culture medium for 2 hours, which demonstrated that whether the surface is treated with integrin $\alpha\beta3$ has little effect on the amount of protein adsorption. In addition, the cells added to the surface in a serum-free medium, as the original design of our study in Figure 5e-g of our manuscript, may not have begun to secrete a large amount of protein during the cell adhesion period within 2 hours. Therefore, integrin $\alpha\beta3$ -blocking has little effect on the quantity of adsorbed proteins.

Supplementary Figure 8. Quantification of adsorbed serum protein on DM₅₀CO₅₀ and RGD modified surfaces that were incubated with serum-containing media first and then treated with or without integrin $\alpha\beta3$ before cell seeding. Statistical analysis: one-way ANOVA with Tukey post-test, * $p < 0.05$

5. *What about the many other, more abundant proteins in serum? I would assume they are equally likely to adsorb to these polymers via hydrophobic or ionic interactions.*

Response: We thank the reviewer for this question. Above discussion in our response to Q2 showed that the surface adsorbed total protein after 2 h incubation of serum-containing medium is at about 11.1 ng/well. The sum of three individual protein (FN, VN, COLL) is at about 1.4 ng/well after cell culture on the surface for 2 days; however, the sum of specific proteins (FN, VN, COLL) after 2 h incubation of serum-containing medium is below the detecting limit of our test and logically is lower than the quantity of adsorbed protein after 2 days. This conclusion encouraged us to further explore the surface adsorption of bovine serum albumin (BSA) using a pure FITC-labeled BSA. We found that BSA has significant adsorption on our polymer surface compared to RGD and KRSR modified surfaces, as shown in the figure below. This conclusion is consistent with the reviewer's prediction. Since BSA is not very relevant to surface cell adhesion, we didn't include BSA as one of our exploration targets.

Fluorescent intensity of BSA-FITC adsorption on variable surfaces

6. *The surface characterization data are important and at least a synopsis of these should be brought into the main body of the paper and some discussion given. More discussion of the cell adhesion and protein adsorption results should be given in relation to these data.*

Response: We thank the reviewer for this suggestion that helps us improved our manuscript with new data and extra discussion. Frist, we added X-ray photoelectron spectroscopy (XPS) characterization to the main text (Figure 1i in our revised manuscript) as shown below. XPS characterization showed that apparent C1s and N1s peaks appeared in surface amination step, which indicated a successful surface functionalization with amine groups by using (3-aminopropyl) triethoxysilane (APTES). A subsequent modification with OEG8 antifouling layer and further DM₅₀CO₅₀ functionalization were confirmed by C:N element ratio change (6.5 for NH₂, 8.2 for OEG8, and 6.8 for DM₅₀CO₅₀).

Secondly, we further characterized the surface topography using Atomic Force Microscopy (AFM). Representative AFM results are shown below (Figure 1j in our revised manuscript). 3D images show that the bare glass surface is quite smooth with $R_a = 0.16$ nm (R_a , a value reflects the surface roughness). Surfaces modified with either RGD or DM₅₀CO₅₀ have increased R_a values, $R_a = 0.73$ nm and $R_a = 0.57$ nm respectively, which indicated successful surface modification. It's noteworthy that DM₅₀CO₅₀ has homogeneous modification on the surface, which is favorable to generate reproducible and homogenous protein adsorption and subsequent cell adhesion.

Moreover, we put the water contact angle data into the main text (Figure 1k in our revised manuscript) as shown below. When the glass surface was modified with OEG8, the surface turns to be more hydrophilic with water contact angle changing from 57° (bare glass surface) to 44° (OEG8 surface). When the OEG surface was furthered modified with our cationic-hydrophobic amphiphilic polymer DM₅₀CO₅₀, the polymer modified surface has a water contact angel of 57°, which reflected the amphiphilic characteristic of the cell adhesive polymer. Such amphiphilic and cationic surface, as we proposed for the design of cell adhesive β -amino acid polymer, facilities surface adsorption of serum proteins that further support cell adhesion on the polymer-modified surface.

Fig. 1 | **i**, XPS spectra of bare glass, NH₂-glass, OEG8 and DM₅₀CO₅₀ modified surface. **j**, AFM images of bare glass, RGD and DM₅₀CO₅₀ modified surface. **k**, Water contact angle of bare glass, OEG8, RGD and DM₅₀CO₅₀ modified surface.

We analyzed the amount of surface adsorbed total protein after variable surfaces were incubated with a serum-containing cell culture medium for 2 hours. Compared to RGD modified surface, DM₅₀CO₅₀ modified surface adsorbed significantly higher amount of serum protein as shown in the figure below (Figure 5a in our revised manuscript). Moreover, we characterized the morphology of serum protein adsorbed on DM₅₀CO₅₀ and RGD modified surface using AFM (Figure 5b in our revised manuscript). We observed more and larger protein aggregates on the DM₅₀CO₅₀ modified surface than that on the RGD modified surface, which may be caused by the ability of the polymer surface to adsorb more serum protein than does the RGD modified surface. We also did AFM characterization on the surface adsorption of important cell-adhesive single serum protein, such as FN, after surface incubation with the protein for 2 hours, and we observed a similar result (Figure 5c in our revised manuscript). On the DM₅₀CO₅₀ modified surface, the abundant and homogeneous distribution of adsorbed proteins echoed the evenly adhered cells to the surface.

All above relevant discussions have been added into our revised manuscript.

Fig. 5 | **a**, The amount of surface adsorbed total protein after variable surfaces were incubated with a serum-containing cell culture medium for 2 hours. **b**, **c** AFM images of DM₅₀CO₅₀ and RGD modified surfaces after incubation with serum-containing cell culture media (**b**) and pure fibronectin (**c**) for 2

hours.

7. *The acronyms for the subunits: CP, CH, CO, NM, MM, DM, HLys should be explained.*

Response: We thank the reviewer for this suggestion and we have added the full name of those racemic subunits in our manuscript, based on the sidechain group, as “CP (cyclopentyl), CH (cyclohexyl), CO (cyclooctyl), NM (“no methyl”), MM (“monomethyl”), DM (“dimethyl”), or HLys (homo-lysine)”.

8. *The English is generally very good but a few “Sinocisms” persist and should be corrected, especially the frequent absence (or incorrect presence) of the plural “s”.*

Response: We thank the reviewer for pointing out this. We have carefully gone through the manuscript and made changes accordingly.

For example: “...using preosteoblasts cell as a model...” was changed to “...using preosteoblast cell as a model...”, and “...the cell adhesion function and mechanism of RGD and KRSR peptides...” was changed to “...the cell adhesion functions and mechanisms of RGD and KRSR peptides...”.

Reviewer #2 (Remarks to the Author):

In this manuscript Chen Q. et al present an interesting work on the use of cationic-hydrophobic amphiphilic β -amino acid polymers to generate new cell adhesive materials. The work presented here may have the potential to develop novel cell adhesive materials that resist to protease, cheap and easy to prepare in large quantity. The in vitro results obtained on pre-osteoblastic cells are convincing and the β -amino acid polymers give significantly higher adhesion by comparison to classically used adhesive peptides such as RGD and KRSR. However, some experiments are wrongly described and more importantly, some experiments are insufficiently interpreted. In addition, the in vivo results are less convincing than in vitro results. More widely, the paper is missing a correct discussion of the presented results.

Response: We thank the reviewer for the comments and specific questions below.

1) *From their Figure 2 the authors assert that their DM50CO50 polymer implies a different cell adhesion mechanism from that of RGD and KRSR peptides. It appears rather that the adhesion is similar to RGD peptide but delayed in time. The mechanism cannot be concluded from this study. Moreover, the authors did not discuss the cell spreading which appears lower on DM50CO50 than on RGD peptide notably after 24h. However, for concluding if there is a real difference in term of spreading and if it's not an effect of cell density, the area of individual cells must be quantified and compared between the surfaces at 3 and 24 hours.*

Response: We thank the reviewer for the comment that inspired us to provide more clear explanation on the cell spreading and hypothesis on cell adhesion mechanism. We made the statement in our revised manuscript that “3 hours after cell seeding, cell spreading area on DM₅₀CO₅₀ surface

and KRSR surface was lower than that on RGD surface; after 24 hours, cell spreading area on DM₅₀CO₅₀ surface and RGD surface are comparable, and both are significantly better than cell spreading on KRSR surface.” This description is supported by the quantification of the cell area after cell seeding for 3 hours and 24 hours, as shown in the figure below (Supplementary Fig. 5 in our revised manuscript).

Supplementary Figure 5. Cell area of preosteoblast cells grown on RGD, KRSR and DM₅₀CO₅₀ modified surfaces for 3h (a) and 24 h (b), respectively. Statistical analysis: one-way ANOVA with Tukey post-test, * $p < 0.05$

It is known that RGD leads to cell adhesion and spreading via direct binding to cell surface integrin; whereas, KRSR leads to cell adhesion and spreading via binding to cell surface heparin. The difference in cell adhesion mechanism between RGD and KRSR explains the difference in cell spreading and cell area. The polymer DM₅₀CO₅₀ supported cell adhesion and spreading different from both RGD and KRSR, and we therefore hypothesize that DM₅₀CO₅₀ have a cell adhesion and spreading mechanism different from either RGD or KRSR. This hypothesis was confirmed by our subsequent mechanism study that DM₅₀CO₅₀ supports cell adhesion by two mechanisms mainly, both an integrin-dependent mode involving binding integrin via adsorbed ECM proteins and a mode involving direct binding of the cationic polymer to cell surface anionic polysaccharides. It takes some time for initial protein adsorption on DM₅₀CO₅₀ surface and it is reasonable to observe a delayed cell adhesion on DM₅₀CO₅₀ surface compared to RGD surface, and a comparable cell adhesion and spreading on these two surfaces after 24 hours. We added related discussion in our revised manuscript.

2) *The RNA-seq analysis is really interesting and give important results about the difference between adhesive peptides and DM50CO50 polymer's influence on cells. The most differences concern the downstream effect of ECM reorganization as highlighted by the authors. However, other GO terms are not discussed by the authors while their -lgP is relatively high (Fig 3b). In particular, several GO terms relative to cell motility or cell division appear in high position in biological processes compared to ECM reorganization and must be also discussed. Indeed, after influencing their adhesion, the DM50CO50 polymer could also influence strongly the migration capacity of preosteoblastic cells and their capacity to colonize new areas during bone regeneration. In addition, the same can be seen in the KEGG pathway enrichment results with the highest score for the ECM-interaction pathway. However, the Wnt signalling pathway like focal adhesion pathway are also present with a relatively high -lgP (Fig. 3c). These two pathways have been shown to be involved in cell migration and cell proliferation. In conclusion, a deeper discussion of this RNA-seq analysis is missing in this paper.*

Response: We thank the reviewer for the suggestive comment that inspired us to have further analysis on the RNA-seq data and strengthen our paper. We reanalyzed the top 10 key GO terms in

biological processes (Figure 3b in our revised manuscript) and found that in addition to two GO terms related to ECM reorganization, four of the top 10 key GO terms are related to cell motility and cell migration (positive regulation of cell migration (GO:0030335), positive regulation of cell motility (GO:2000147), positive regulation of cellular component movement (GO:0051272) and muscle cell migration (GO:0014812)). The other four of the top 10 key GO terms are related to cell division (regulation of nuclear division (GO:0051783), regulation of mitotic nuclear division (GO:0007088), chromosome segregation (GO:0007059) and mitotic nuclear division (GO:0140014)). These results imply that DM₅₀CO₅₀ polymer could influence cell adhesion as well as the migration capacity of preosteoblast cells and their capacity to colonize new areas during bone regeneration, which is supported by the favorable result for *in vivo* bone regeneration study in a rat cranial defect model. In the KEGG pathway enrichment analysis we also observed the highest score for the ECM-interaction pathway, which is consistent to the result in GO analysis above. We also agree with the reviewer that two other pathways, the Wnt signaling pathway and focal adhesion pathway, are also enriched. We paid attention to these two pathways that are known to involve in cell migration and cell proliferation, with focal adhesion involving in reorganization of the actin cytoskeleton and a prerequisite for changes in cell shape and motility, and Wnt signaling pathway involving in remodelling of the cytoskeleton and changes in cell adhesion and motility. This result suggested that DM₅₀CO₅₀ polymer may not only supported the cell adhesion and spreading but also affect the cell mobility and proliferation, which is important for cell recruitment and the formation of new bone during bone repair. Related discussions above were added to our revised manuscript.

Fig. 3 | RNA-seq analysis on DM₅₀CO₅₀ modified surface compared to RGD modified surface.

3) In the further parts, the authors have smartly demonstrated that in a serum-containing environment, DM50CO50 promoted cell adhesion by partially following an integrin-dependent mechanism whereas,

in a serum-free environment, DM50CO50 promoted cell adhesion by following an integrin-independent mechanism, possibly via direct binding to cell surface polysaccharides such as heparin. These results were confirmed using an anti $\alpha\beta3$ integrin antibody or by degrading cell surface polysaccharides. Therefore, it could be interesting to discuss these results with the RNA-seq results and the different highlighted signal transduction pathways.

Response: We thank the reviewer for the suggestive comment that inspired us to correlate our experimental study in mechanism with the RNA-seq results and different highlighted signal transduction pathways. In a serum-containing environment, our study indicated that DM₅₀CO₅₀ promoted cell adhesion by following an integrin-dependent mechanism via surface adsorbed proteins. Quantification on surface adsorbed individual protein (FN, VN, Coll and LAM) on MC3T3-E1 cultured surface after 2 days incubation in a serum-containing environment showed that FN and Coll are the most abundant proteins on DM₅₀CO₅₀ modified surface, and the amount of adsorbed FN and Coll on DM₅₀CO₅₀ modified surface is also significantly higher than that on the RGD modified surface as shown in the figure below (Figure 5d in our revised manuscript). According to gene ontology (GO) enrichment analysis in RNA-seq we found that both fibronectin binding (GO:0001968) and collagen binding (GO:0005518) terms are in the top three of the molecular function categories, which implies a conclusion the same as above experimental study on cell adhesion mechanism in serum-containing environment that fibronectin and collagen are two important proteins involved when the polymer promotes the adhesion of preosteoblast cells. Related discussion was added to our revised manuscript.

In a serum-free environment, our study indicated that DM₅₀CO₅₀ promoted cell adhesion by following an integrin-independent mechanism possibly via direct binding to cell surface polysaccharides such as heparin as supported by the result in $\alpha\beta3$ integrin blocking study and the result in cell surface polysaccharides degradation study. In our RNA-seq analysis on cells after incubation for 2 days in a serum-containing environment as shown in the figure below (Figure 3b in revised manuscript), the direct interaction between DM₅₀CO₅₀ and cell surface polysaccharides is diminished by the polymer surface adsorbed proteins, nevertheless, in the GO enrichment analysis we still observed glycosaminoglycan binding (GO: 0005539) term as one of the top molecular function categories, which implies a conclusion the same as above experimental study on cell adhesion mechanism in serum-free environment that polysaccharides, such as heparin, are involved when the polymer promotes the adhesion of preosteoblast cells. Related discussion was added to our revised manuscript.

Fig. 5 | d, Proteins density on DM₅₀CO₅₀ and RGD modified surfaces after cells were cultured on the surfaces for 2 days. Statistical analysis: t-test, ** $p < 0.001$, *** $p < 0.0001$.

Fig. 3 | b, Top 15 most significant gene ontology (GO) terms of associated molecular functions from differentially regulated genes (P represents the significance level of enrichment).

4) An anti $\alpha\beta3$ antibody was used to block the interaction of the $\alpha\beta3$ integrin with the RGD peptide in cells adhering on surfaces modified with RGD peptide or the DM50CO50 polymer. However, the description of this experiment is unclear as well in the results part as in the materials and method part showing that this experience was not well understood by the writer, casting doubt on its validity. Why the antibody against $\alpha\beta3$ integrin is incubated with the modified surfaces before seeding the cells? What will the antibody interact with? Actually, the correct experience would be to treat the cells with the anti $\alpha\beta3$ integrin antibody before to seed them on the different surfaces. In the materials and method part, a better title would be “Antibody anti $\alpha\beta3$ integrin to block cell adhesion through RGD” instead of “Antibody $\alpha\beta3$ to block RGD”. The authors must rewrite the materials and method as well as the results parts to make this experiment clearer and convince the reader of its validity.

Response: We thank the reviewer for pointing out this important issue in the footnote of Figure 5 in our original manuscript, where we wrote "integrin $\alpha\beta3$ " as " $\alpha\beta3$ antibody" by mistake (in the main text, we used the correct term "integrin $\alpha\beta3$ "). In fact, we used recombinant human integrin $\alpha\beta3$ instead of anti- $\alpha\beta3$ antibody in our experiment by following a precedent literature (*J. Am. Chem. Soc.* 2016, 138, 15078-15086). In our experiment, by adding integrin $\alpha\beta3$ to the serum-incubated material surface, integrin $\alpha\beta3$ can interact directly with the RGD motif (either from the RGD motif of surface adsorbed proteins or the RGD peptide) on the material surface to block these RGD from binding with cell surface integrin. After two hours, we observed that $\alpha\beta3$ attenuated cell adhesion to DM₅₀CO₅₀ modified surface by reducing cell spread area, but without affecting adhered cell density; whereas, $\alpha\beta3$ substantially attenuated cell adhesion to RGD modified surface on both spread area and adhered cell density. This result indicated that cell adhesive β -amino acid polymers can exploit adsorbed proteins and expose their RGD motif for direct binding with cell surface integrin, whereas, the unaffected cell density on $\alpha\beta3$ treated DM₅₀CO₅₀ surface indicated that cell adhesive β -amino acid polymers have additional cell adhesion mechanisms as supported by subsequent study of our manuscript.

Fig. 5 | e,f,g. Fluorescence micrographs (e), spread area (f) and density analysis (g) of cells 2 hours after seeding on DM₅₀CO₅₀ and RGD modified surfaces that were incubated with serum-containing media first and then treated with or without integrin $\alpha\beta3$ before cell seeding.

In addition, we also took the reviewer's suggestion and did the suggested experiment. This suggested study using anti-integrin $\alpha\beta3$ antibody to treat cell first and attempt to block cell surface integrin, as a parallel approach to our design mentioned above. We treated cells with the anti-integrin $\alpha\beta3$ antibody before cell seeding on surfaces that are incubated with serum-containing medium. After 2 hours, observation on cell adhesion showed no significant difference in the amount of cell adhesion and spreading area for cells with or without anti-integrin $\alpha\beta3$ antibody treatment. It is known that RGD can bind to not only integrin $\alpha\beta3$ but also other types of integrin, such as $\alpha\beta5$, $\alpha\beta1$, etc. (*Chem. Rev.* **2017**, 117, 14015–14041; *Adv. Mater.* **2020**, 32, 1902604). Therefore, only blocking the integrin $\alpha\beta3$ on the cell membrane does not prevent RGD from binding with other integrins and promoting cell adhesion, as we observed in the suggested experiment by the reviewer and shown in the figure below.

5) *The in vivo demonstration of the efficiency of DM50CO50 to promote bone formation is not really convincing since the bone formation increases from 5% with the control material (PEG) to 10% of the total volume of the cavity which is still 90% empty. Therefore, the authors' assertion that their scaffold greatly promotes in vivo bone regeneration is really exaggerated and needs to be revised.*

Response: We thank the reviewer for the comment that inspired us to carefully analyze our *in vivo* experiment protocol. In our original experiment using DM₅₀CO₅₀ modified PEG hydrogel that contains 20 wt% of PEG, we found that the majority of the hydrogel was not degraded when we collected the data after 8 weeks of implantation, which implies an unfavorable invasion of new bone to the hydrogel scaffold. Therefore, in our revision we optimize DM₅₀CO₅₀ modified PEG hydrogel by reducing PEG content from 20 wt% to 5 wt% to facilitate a favorable invasion of new bone. Moreover, in the new *in vivo* experiment we also included two commercial bone repair materials for comparison, Gelatin methacryloyl (GelMA) and polylactic acid (PLA). GelMA possesses RGD sequence and has been widely used in various biomedical applications including bone regeneration (*Biomaterials* **2015**, 73 254-271). PLA has been widely used clinically for bone regeneration (*Advanced Drug Delivery Reviews* **2016**, 10, 7247–276). According to our new *in vivo* experimental results of Micro-CT and Masson's trichrome staining analysis after implantation for 8 weeks, as shown in the figure below (Figure 7 in our revised manuscript), DM₅₀CO₅₀ functionalized PEG hydrogels perform better than the bare PEG hydrogel group, GelMA group and PLA group in

promoting bone regeneration.

Fig. 7 | **b**, Comparison before and after PEG hydrogel swelling. **c**, Preosteoblast cells adhesion to bare PEG hydrogel and DM₅₀CO₅₀-modified PEG hydrogel after cell seeding for 2 days. **d,e,f**, Masson's trichrome stain (**d**), 3D histomorphometric analysis of Micro-CT (**e**) and bone volume analysis (**f**) of the cranial bone samples from bare PEG hydrogel, DM₅₀CO₅₀-modified PEG hydrogels, GelMA and PLA after 8 weeks of implantation. Statistical analysis: t-test, **p* < 0.05.

Minor points:

- The preosteoblastic cells used by the authors are referred in the Materials and methods as MC3T3 cells. The sub-line must be given. Classically, it's the sub-line E1 that is used.

Response: We thank the reviewer for pointing out this and we have changed "MC3T3" to "MC3T3-E1" in our manuscript.

- Fig. 5 caption: the authors should choose between protein density or protein adsorption. A density is generally expressed in function of the area. Can the technique used really quantify the protein density?

Response: We thank the reviewer for this suggestion. In response to reviewer 1 comment, we quantified the surface adsorbed protein and updated the Figure 5c-d as shown below. For the new data of quantified protein adsorption we described as “the amount of” surface adsorbed protein.

Figure 5. c, The amount of surface adsorbed total protein after variable surfaces were incubated with a serum-containing cell culture medium for 2 hours. d, The amount of surface adsorbed four protein (FN, VN, COLL, LAM) on DM₅₀CO₅₀ and RGD modified surfaces after cells were cultured on the surfaces for 2 days

- *Materials and methods part on hydrogel preparation: the authors describe a silicon mould (and not mode) with 3 mm in diameter while the critical size cranial defect was drilled using a 5 mm diameter trephine. What is the correct diameter size for the hydrogel used in the in vivo experiment?*

Response: We thank the reviewer for pointing out this important issue that reminds us to clarify our operation to readers. Hydrogels were made in a silicon mode with a 3 mm diameter and then these hydrogels were swollen in PBS to remove unreacted free molecules. Then we obtained the hydrogel at a final diameter at 5 mm, which matches the critical size cranial defect. To make this procedure clear to the reviewer and readers, we added photos of hydrogels before and after swollen step as shown blow (Figure 7b in our revised manuscript).

Fig. 7 | b, Comparison before and after PEG hydrogel swelling.

- *Materials and methods part on histological analysis: “After microangiography analysis, tissue samples were subjected to histological analysis”. What is this microangiography analysis?*

Response: We thank the reviewer for pointing out this. We used the term “microangiography analysis” by mistake. We have changed it to the correct term of “Micro-CT evaluation”.

We hope that the revised manuscript will prove to be acceptable for publication in *Nature Communications*.

Thank you for your assistance.

Sincerely,

Runhui Liu

Professor of Chemistry and Biomaterials

REVIEWERS' COMMENTS

Reviewer #1 (Remarks to the Author):

The authors have answered my comments in detail and in a satisfactory way. They have revised the manuscript appropriately. I recommend acceptance of this version.

John L. Brash

Reviewer #2 (Remarks to the Author):

I would like to thank the authors for responding well and taking into account all the reviewers' comments. I note that the manuscript is now much improved. I consider that the manuscript could be published in Nature communications after a few small corrections have been made.

- A reference should be added to justify the relevance of the protocol used by the authors to block the adhesion of cells on RGD peptides by adding alpha-v beta-3 integrin to the surfaces covered by the DM50CO50 polymer or by RGD peptides. This is not a common protocol and this protocol is actually not described in the article given as reference by the authors in their response to my comment (ref 24). This reference should be added in the manuscript in the corresponding material and method section.

- The English should be revised

Reviewer #1 (Remarks to the Author):

The authors have answered my comments in detail and in a satisfactory way. They have revised the manuscript appropriately. I recommend acceptance of this version.

John L. Brash

Response: We thank the reviewer, Prof. Brash, for the inspiring comments and questions during the peer review process to help us substantially improve our manuscript.

Reviewer #2 (Remarks to the Author):

I would like to thank the authors for responding well and taking into account all the reviewers' comments. I note that the manuscript is now much improved. I consider that the manuscript could be published in Nature communications after a few small corrections have been made.

Response: We thank the reviewer for the inspiring comments and questions during the peer review process to help us substantially improve our manuscript and the revision suggestion listed below.

- A reference should be added to justify the relevance of the protocol used by the authors to block the adhesion of cells on RGD peptides by adding alpha-v beta-3 integrin to the surfaces covered by the DM50CO50 polymer or by RGD peptides. This is not a common protocol and this protocol is actually not described in the article given as reference by the authors in their response to my comment (ref 24). This reference should be added in the manuscript in the corresponding material and method section.

Response: We thank the reviewer for the kind reminder. The protocol is described in the supporting information of the reference (ref 24 as indicated in our previous one-on-one response) that “For RGD peptide, the TiO₂-coated quartz substrate was first incubated with 250 μL of a commercial human integrin αβ₃ (10 μg/mL in PBS, CF, R&D, USA). After incubation at 37 °C for 1 h, the surfaces were washed with PBS for three times.” We have added this reference (ref 24) into our manuscript in the corresponding method section as below.

“...incubated with integrin αβ₃ (10 μg/mL in PBS) at 37 °C for 1 h²⁴...”

- The English should be revised

Response: We thank the reviewer for the suggestion. We have further revised our English throughout the manuscript.